# RoboMD: Uncovering Robot Vulnerabilities through Semantic Potential Fields

**Som Sagar**
Arizona State University
`ssagar6@asu.edu`

**Jiafei Duan**
University of Washington

**Sreevishakh Vasudevan**
Arizona State University

**Yifan Zhou**
Arizona State University

**Heni Ben Amor**
Arizona State University

**Dieter Fox**
University of Washington

**Ransalu Senanayake**
Arizona State University

## Abstract

Robot manipulation policies, while central to the promise of physical AI, are highly vulnerable in the presence of external *variations* in the real-world. Diagnosing these vulnerabilities is hindered by two key challenges: (i) the relevant variations to test against are often unknown, and (ii) direct testing in the real world is costly and unsafe. We introduce a framework that tackles both issues by learning a separate deep reinforcement learning (deep RL) policy for vulnerability prediction through virtual runs on a continuous vision–language embedding trained with limited success-failure data. By treating this embedding space, which is rich in semantic and visual variations, as a potential field, the policy learns to move toward vulnerable regions while being repelled from success regions. This vulnerability prediction policy, trained on virtual rollouts, enables scalable and safe vulnerability analysis without expensive physical trials. By querying this policy, our framework builds a probabilistic vulnerability-likelihood map. Experiments across simulation benchmarks and a physical robot arm show that our framework uncovers up to 23% more unique vulnerabilities than state-of-the-art vision–language baselines, revealing subtle vulnerabilities overlooked by heuristic testing. Additionally, we show that fine-tuning the manipulation policy with vulnerabilities discovered by our framework improves performance with much less data. GitHub: https://github.com/somsagar07/RoboMD.

## 1 Introduction

Learning robust robot manipulation policies is widely regarded as the foundational problem in physical AI. A robust solution would unlock capabilities ranging from reliable industrial automation in cluttered factory environments to assistive humanoid arms that seamlessly interact with people in everyday settings. In perception and language tasks, vulnerabilities can often be quickly identified by querying large datasets or benchmarks, with little cost beyond computation. In manipulation, however, discovering vulnerabilities is far more difficult: it requires physical trials that are slow, expensive, and potentially unsafe, posing risks not only to the robot but also to the environment and even people. This makes naive heuristic testing or trial-and-error both impractical and costly, motivating the need for scalable, active vulnerability exploration methods pre-deployment. In this paper, we pursue this by *training a separate vulnerability prediction model*, formulated as a deep reinforcement learning (deep RL) policy that actively searches for failures.

Yet, even with a scalable search framework, a second obstacle remains: *what exactly should we test for?* Manipulation policies must withstand diverse and unpredictable *variations*. For instance, referring to Fig. 1, a robot designed to grasp a bottle should generalize across various colors, shapes, sizes, and materials, and remain effective under changes in lighting, background, and physical layouts Pumacay et al. (2024); Xie et al. (2024). Naively applying deep RL to a handful of known

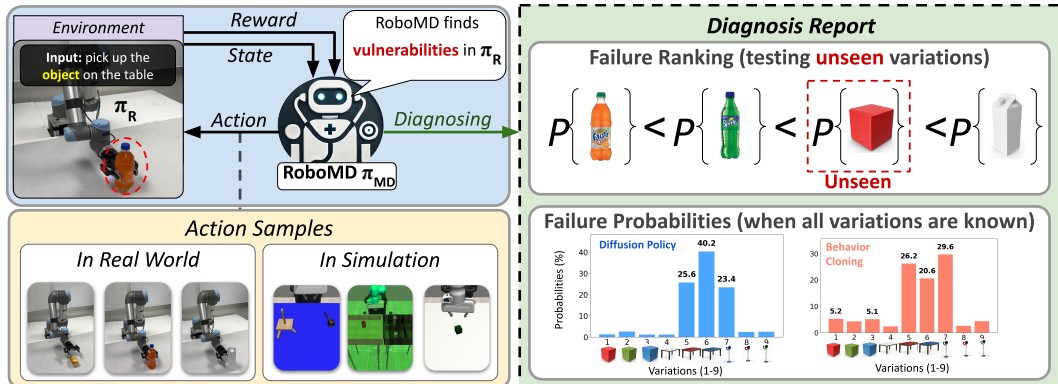

Figure 1: The vulnerability diagnostic pipeline. We train RoboMD ($\pi_{MD}$), a deep reinforcement learning (RL) policy, by probing a pre-trained manipulation policy ($\pi_R$) while systematically applying *environmental variations* (e.g., background colors, object shapes, darkness levels, etc.) as RL actions, examples of which are shown in the "Action Samples" panel (bottom left). Note that these examples are for intuitive understanding only; the user does not need to specify all variations. By observing the success or failure on each trial of $\pi_R$, RoboMD learns to identify vulnerabilities even in unseen variations (e.g., a red cube, in this example). The final output is a diagnosis report that can either provide a ranked list of failure likelihoods, including for previously unseen variations (top right), or quantify the failure probabilities for previously seen variations (bottom right).

variations to search for failures risks overlooking critical failure modes that emerge under novel conditions; Lin et al. (2024).

We overcome these limitations by reformulating deep RL exploration over a learned, continuous vision-language embedding space rather than discrete, hand-specified variations. Constructed from limited success-failure data, we ensure that this embedding space is rich in semantic and visual structure. We cast the learned space as a potential field that guides exploration toward failures and away from successes. This, in turn, allows us to train the deep RL-based vulnerability search policy that organically uncovers relationships and adapts to diverse variations. This policy can be queried at any time to predict whether a scenario will lead to failure, enabling scalable and systematic diagnosis of manipulation policies. The main contributions of the paper can be summarized as:

1. Proposing a deep RL-based framework that operates in a semantically rich embedding space, for diagnosing failures in pre-trained manipulation policies.
2. Providing extensive experimental evidence on simulated and real-world setups, backed by theoretical guarantees.
3. Systematically improving robot policies using the failures diagnosed by our framework.

## 2  RELATED WORK

**Failures in large models** can be characterized by querying vision-language foundation models Agia et al. (2024); Duan et al. (2024); Klein et al. (2024); Subramanyam et al. (2025); Liu et al. (2023) or searching for failures Sagar et al. (2024). As we further verify in experiments, the former does not show strong performance in deciphering failures as they do not iteratively interact with the robot policy. Furthermore, VLM models alone are not yet capable of making highly accurate quantitative predictions such as probabilities, making them difficult to use in high-stakes tasks. In the latter approach, outside of robotics, deep RL has recently been employed in machine learning to identify errors in classification and generation Sagar et al. (2024). Similarly, Delecki et al. (2022); Hong et al. (2024) utilized Markov decision processes to explore challenging rainy conditions, which is backed by the work of Corso et al. (2021), highlighting the role of sequential decision-making models to ensure the safety of black-box systems. Note that these approaches have neither been demonstrated on complex physical systems like manipulation nor can they generalize beyond a fixed set of known failures, both of which we address.

**Out-of-distribution (OOD) detection** methods can also be used to identify unseen inputs, for instance, in automotive perception Nitsch et al. (2021), runtime policy monitoring Agia et al. (2024), and regression Thiagarajan et al. (2023). However, *failure detection constitutes a different prob-*

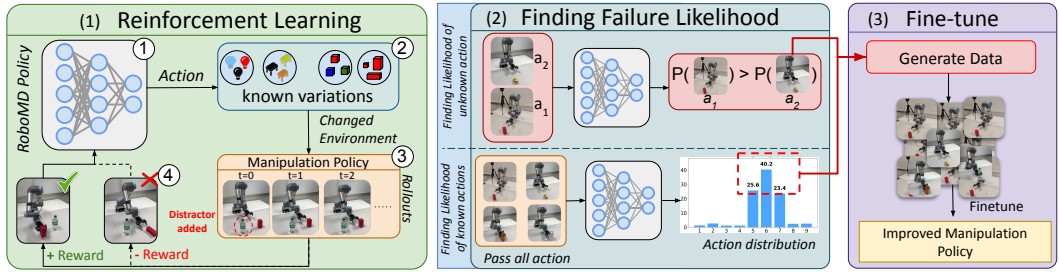

Figure 2: Our framework operates in three stages: (1) a PPO-based deep RL agent ($\pi_{\mathrm{MD}}$) perturbs the environment to reveal configurations that cause failures in the pre-trained manipulation policy ($\pi_{\mathrm{R}}$); (2) its learned action distribution conditioned on the input observation is converted into failure-mode probabilities, either over a continuous embedding for novel changes or via discrete candidates; and (3) those probabilities are used to fine-tune $\pi_{\mathrm{R}}$ for improved performance.

*lem than OOD detection*, as not all OOD samples lead to failure, and failures can also occur in-distribution. We aim to characterize failures both within and beyond the training distribution, not merely flag OOD instances. A related area of research is uncertainty quantification, which underpins many OOD detection methods. While many attempts have been made to characterize the epistemic uncertainty Senanayake (2024), the unknown unknowns, in robot perception systems O'Callaghan & Ramos (2012); Kendall & Gal (2017), only a few attempts have been made to address this challenge in deep RL Jiang et al. (2024) and imitation learning Jeon et al. (2018); Brown et al. (2020); Ramachandran & Amir (2007). As robot policy models grow increasingly complex, formally characterizing epistemic uncertainty becomes extremely challenging. Even if we can, such techniques do not inform engineers where the models fail, making it harder to further improve the policies.

**Generalized policies** are less prone to failures. Toward achieving this goal, generalization in robotics has been extensively studied to enable robots to adapt to diverse scenarios. Large-scale simulation frameworks have been developed to evaluate the robustness of robotic policies across varied tasks and environmental conditions Pumacay et al. (2024); Fang et al. (2025). Vision-language-action models trained on multimodal datasets have demonstrated significant advancements in improving adaptability to real-world scenarios Brohan et al. (2022; 2023). Additionally, approaches such as curriculum learning and domain randomization have proven effective in enhancing generalization by exposing models to progressively complex or randomized environments Andrychowicz et al. (2020). These methodologies collectively address the challenges of policy robustness. In contrast to these training focused methods, our framework acts as a diagnostic tool that is complementary to them as it can systematically identify failures in policies trained using any such approaches. Ultimately, no matter how general the model is, unforeseen conditions and subtle variations will always rise, making systematic diagnostic tools indispensable for real-world deployments. Others have tackled safety from control-theoretic Bajcsy & Fisac (2024); Grimmeisen et al. (2024), human-factors Sanneman & Shah (2022), statistical-certification Farid et al. (2022); Ren & Majumdar (2022); Yang et al. (2020); Vincent et al. (2023), and formal-methods Tmov et al. (2013) perspectives. While these enhance overall robustness, our framework diagnoses failures pre-deployment to help guide policy improvement.

## 3 METHODOLOGY

We analyze vulnerabilities of a *pre-trained* manipulation policy, $\pi_{\mathrm{R}}$, by training a separate vulnerability prediction policy, named RoboMD, $\pi_{\mathrm{MD}}$. RoboMD is designed to be agnostic to the architecture or underlying training method of $\pi_{\mathrm{R}}$. Whether $\pi_{\mathrm{R}}$ is trained via behavioral cloning, reinforcement learning, foundation models, or any future methods, $\pi_{\mathrm{MD}}$ only requires rollouts of $\pi_{\mathrm{R}}$, making $\pi_{\mathrm{MD}}$ adaptable to a wide range of manipulation policies and tasks. In Section 3.1, we first formalize the failure diagnosis problem as a sequential search problem. Building on this formulation, in Sections 3.2 and 3.3, we show how $\pi_{\mathrm{MD}}$ can search for vulnerabilities in space of variations that is not known. We also show a special case of it, where we can systematically test over a discrete set of known variations. After describing how vulnerabilities can be queried at runtime in Section 3.4, we theoretically ground each step in our framework in Section 3.5.

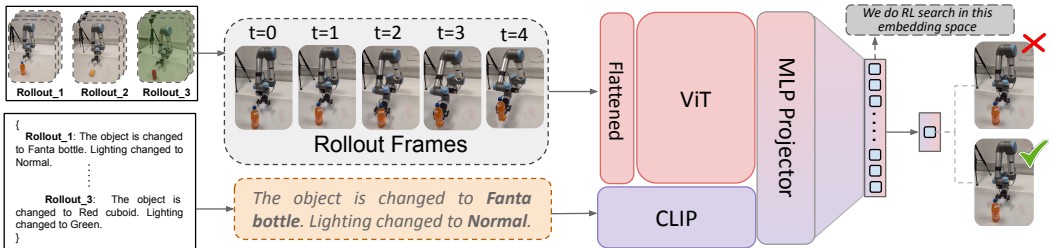

Figure 3: The pipeline shows how rollouts with variations (e.g., object or lighting changes) are processed to learn meaningful embeddings. Visual data and text from the rollouts are embedded using ViT and CLIP, and then projected to an MLP, followed by failure-success classification.

## 3.1 FORMALIZING FAILURE DIAGNOSIS AS A SEARCH PROBLEM IN A SEMANTIC SPACE

> **Overview.** As rationalized in Section 1, we first learn a continuous vision-language embedding to project raw manipulation data into a semantically meaningful representation of the success or failure of $\pi_R$ (Fig. 3). We then learn $\pi_{MD}$ by exploring this space using proximal policy optimization (Fig. 10). Once trained, $\pi_{MD}$ can be queried to predict a ranked map of vulnerable conditions for the manipulation task. An overview of how to train and query $\pi_{MD}$, along with how to use $\pi_{MD}$ to improve $\pi_R$ is shown in Fig. 2.

We consider a continuous semantic embedding space, $\mathcal{E}$, that represents some limited notions of success and failure of $\pi_R$ (details on training this embedding is provided in Section 3.2). We train $\pi_{MD}$ to traverse this space and learn how to predict failures. Treating the manipulation policy $\pi_R(a_t|s_t)$ and the robot's environment as a black box, we formulate this traversal of $\pi_{MD}$ as a Markov Decision Process (MDP), defined by the tuple $\langle \mathcal{S}, \mathcal{A}, \mathcal{P}, R, \gamma \rangle$:

- **State Space** ($\mathcal{S}$): The state $s_t \in \mathcal{S} \equiv \mathcal{E}$ is a realization of the continuous semantic embedding, represented as a vector $s_t \in \mathbb{R}^{512}$. While every physically instantiated environment variation corresponds to a state in $\mathcal{S}$, not all realizations of $\mathcal{S}$ necessarily translate to a physically plausible environment variation. Thus, all $\mathcal{S}$ can be viewed as the set of *hypothesized environment variations*, accessible through the embedding, which may include both physically viable and unviable variations. See Fig. 1 for intuition on variations.
- **Action Space** ($\mathcal{A}$): An action $a_t \in \mathcal{A} \equiv \mathcal{E}$ introduces a variation to $s_t$. By taking different actions, $\pi_{MD}$ can jump from one hypothesized environment variation (a state) to another. While actions can have a physical meaning (e.g., making the environment darker), we do not explicitly define them because an action can be any vector value, $a_t \in \mathbb{R}^{512}$.
- **Transition** ($\mathcal{P}$): The transition function $\mathcal{P}(s_{t+1}|s_t, a_t)$ is determined by applying the variation $a_t$ to the state $s_t$, which results in $s_{t+1}$.
- **Reward** ($R$): $\pi_{MD}$ is rewarded for finding failures quickly.
- **Discount Factor** ($\gamma$): We use a standard discount factor of $\gamma = 0.99$.

The goal of learning $\pi_{MD}$ is to find a sequence of actions (environment variations), which maximizes the probability of $\pi_R$ to fail at its task. The next sections provide exact details of the procedure.

## 3.2 BUILDING THE SEMANTIC EMBEDDING AS A POTENTIAL FIELD OF SUCCESS-FAILURES

To predict vulnerabilities of unseen environments, we need at least two pieces of information: 1) some prior belief of where vulnerabilities might occur and 2) a way to generalize that belief to unseen conditions. We construct the belief from a limited set of labeled rollouts, which are then used to train a vision–language embedding that captures semantic similarity between vulnerabilities. By operating directly in this embedding, $\pi_{MD}$ extends its search to a *continuous action space*.

Our approach hinges on creating this embedding space $\mathcal{E}$ such that the relationship between implicit environment variations and policy failures is locally smooth. More formally, we learn an embedding that acts as a **potential function**, $\Phi$, over the space of environmental variations. Later, in Section 3.5, we show that reward shaping based on a potential difference, $F(s_t, a, s_{t+1}) = \gamma \Phi(s_{t+1}) - \Phi(s_t)$, guarantees that the optimal policy is preserved and provides a dense learning signal for convergence.

To construct this potential field of success-failures, we train a multimodal embedding that organizes environmental variations based on a set of labeled success-failure rollouts. To this end, we collect $M$ rollouts from $\pi_R$ for a given task with $\mathcal{D} = \{(x_i^{\text{vision}}, x_i^{\text{lang}}), y_i\}_{i=1}^M$, where $x^{\text{vision}}$ is the raw image input that we typically provide to manipulation policies, $x^{\text{lang}}$ is a short textual description of the task, and $y \in \{\text{failure, success}\}$. Since we know the action (environment variation) we apply, the textual description can be automatically constructed (see Appendix F). Note that this dataset is created by collecting environmental variations (e.g., object color, lighting changes) chosen based on prior observations, assumptions and knowledge of conditions that often expose policy vulnerabilities. Using this data, as shown in Fig. 3,we train a dual backbone architecture that consists of:

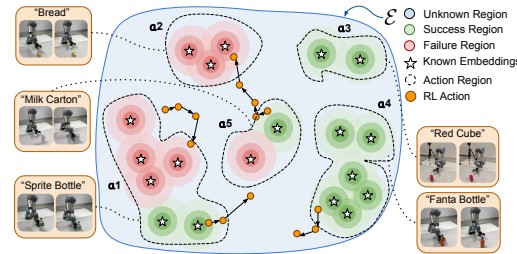

Figure 4: RL exploration in continuous embedding space $\mathcal{E}$. Stars ($\star$) denote known embeddings ($\mathcal{E}_{\text{known}}$) obtained from the dataset $\mathcal{D}$. This induces regions of failure and success. The rest are unknown regions. Dashed boundaries group similar environment variations sharing a common physical meaning. The orange circles and arrows show the RL agent's transition sequence, attracting towards failures and repelling from successes. Since each step is obtained without robot rollouts, each one is a *hypothesized environment variation*. Since the transitions are biased toward failures, $\pi_{\text{MD}}$ learns to encode the failure distribution.

1. A Vision Transformer (ViT) backbone Dosovitskiy et al. (2020) to convert $x_i^{\text{vision}}$ to visual features. Vision transformers capture the contextual relationships within the visual input. Since the original ViT is trained on millions of images from ImageNet with thousands of object categories, this backbone helps to build semantic relationships of everyday environments, so that $\pi_{\text{MD}}$ can infer about an unseen environment from similar environments.

2. A CLIP encoder Radford et al. (2021) to process semantic descriptions. Since robots operate in complex environments, we empirically found (Section 4.2) that providing a task description in natural language helps to focus on the necessary features of the vision input.

The dual backbone combines complementary strengths, resulting in a multimodal embedding that enables better generalization across different environmental variations. The outputs are projected and concatenated, which is passed through a 512-unit MLP layer followed by a classification head. We define the output of the 512 embedding vector as $e_i$ for an arbitrary input $(x_i^{\text{vision}}, x_i^{\text{lang}})$.

The architecture is trained with backpropagation using a joint objective of binary cross-entropy with the classification output $y$ and a contrastive loss objective $\mathcal{L}$ at the MLP layer, which, as we later show in Section 3.5, structures $\mathcal{E}$ as a potential field to enforce smoothness within $\mathcal{E}$. It minimizes the potential difference between embeddings of rollouts with similar outcomes while maximizing it for rollouts with different outcomes. $\mathcal{L}$ is defined as:

$$\mathcal{L} = \sum_{i,j \in \mathcal{D}} \left[ \mathbb{1}_{y_i = y_j} d_{ij} + \mathbb{1}_{y_i \neq y_j} \max(0, m - d_{ij}) \right], \tag{1}$$

where, $d_{ij} = \|e_i - e_j\|_2$ is the Euclidean distance between two randomly sampled points in $\mathcal{D}$, with $m$ hyperparameter margin. The MLP layer projects raw vision-language data to an embedding space, $\mathcal{E}$. The inputs in $\mathcal{D}$ are now assigned to an environmental variation in the semantic space. Because the space is structured such that proximity corresponds to outcome similarity, $\pi_{\text{MD}}$ can now be trained with virtual rollouts to efficiently map the entire landscape of potential failures in $\mathcal{E}$.

## 3.3 DEEP REINFORCEMENT LEARNING IN THE CONTINUOUS SEMANTIC SPACE

The agent $\pi_{\text{MD}}$ can now navigate through $\mathcal{E}$ guided by some of its *known realizations*, $\mathcal{E}_{\text{known}} = \{e_i; i \in \mathcal{D}\}$, a set of pre-computed embeddings derived from $\mathcal{D}$. Note that $\mathcal{E}_{\text{known}} \subset \mathcal{E}$ and $e_i \in \mathbb{R}^{512}$. These embeddings serve as reference points in the action space, representing well-understood regions where failure/success is already observed. As shown in Algorithm 1, the $\pi_{\text{MD}}$ samples an action $a_{t+1}^*$ from the embeddings space and finds the closest embedding in $\mathcal{E}_{\text{known}}$ to obtain its corresponding action $a$, thus performing an action implicitly applies a variation to the environment,

although we are not explicitly changing the physical environment. Therefore, these actions are extremely cheap compared to explicitly changing the environment and running rollouts. To optimize $\pi_{\text{MD}}$ we use PPO Schulman et al. (2017), which provides stable training while maintaining sufficient exploration. In a continuous space, PPO's entropy regularization is especially important, as it prevents the agent from collapsing to a few modes and instead encourages coverage of the broader space. We define the reward function to encourage discovering failure regions while discouraging both large deviations from $\mathcal{E}_{\text{known}}$ and repetitive actions since large deviations lead to uncertain regions, while repetitive actions can indicate a stalled search. This reward mechanism is captured by:

$$R(s, a) = \begin{cases} \frac{K_{\text{failure}}}{\text{penalty}+1} - k \cdot \mathcal{N}(a), & \text{if failure}, \\ -\frac{K_{\text{success}}}{\text{horizon} \times (\text{penalty}+1)}, & \text{if success}. \end{cases} \tag{2}$$

Here, failure-success is inferred from the closest point in $\mathcal{D}$ and the distance penalty in the denominator, scales with $\|a - e\|, \forall e \in \mathcal{E}_{\text{known}}$. The penalty can be related to the potential field in terms of $\|a - e\|_2 = \|\Phi(s_a) - \Phi(s_e)\|_2$, where $s_a$ and $s_e$ are the states reached by applying actions $a$ and $e$, respectively. The frequency penalty, $\mathcal{N}(a)$, counts consecutive repeats of $a$, and we set the coefficient $k = 5$. The frequency penalty serves as a practical mechanism to promote exploration, preventing the agent from repeatedly sampling the same point and pushing it toward uncertain regions. This approach aligns with the principles of Theorem 2, which states that concentrating exploration near the decision boundary between success and failure

---

**Algorithm 1** Learning $\pi_{\text{MD}}$ policy

1: **Inputs:** $\mathcal{D} = \{(x_i^{\text{vision}}, x_i^{\text{lang}}), y_i\}_{i=1}^M$
2: **Precompute:** $\mathcal{E}_{\text{known}}$ from $\mathcal{D}$
3: **Init:** steps $N$, reward $r = 0$, $\pi_{\text{MD}} = \text{rand}$
4: **for** $t = 0$ to $N$ **do**
5:     Sample $a_{t+1} \sim \pi_{\text{MD}}(s_t)$
6:     $a_{t+1}^* \leftarrow \arg\min_{e \in \mathcal{E}_{\text{known}}} \|a_{t+1} - e\|_2$
7:     $s_{t+1} \leftarrow \pi_{\text{MD}}(s_t | a_{t+1}^*)$
8:     $r \leftarrow r + R(s_{t+1})$
9:     **if** failure detected **then**
10:         Reset; $r \leftarrow 0$
11:     **end if**
12: **end for**
13: **Outputs:** RoboMD policy $\pi_{\text{MD}}$

---

leads to more efficient identification of vulnerabilities. Fig. 4 illustrates this process, where RL samples embeddings to steer toward failure-prone regions without requiring full policy rollouts.

**Special case: when the candidate variations are known.** If the set of candidate variations in $\mathcal{E}_{\text{known}}$ is explicitly given (e.g., constructed from historical failures or expert knowledge), then $\pi_{\text{MD}}$ merely has to search over $\mathcal{A} \equiv \mathcal{E}_{\text{known}}$. In this setting, $\pi_{\text{MD}}$ gradually modifies the environment by applying a finite sequence of predefined actions $(a_1, a_2, \ldots, a_n)$ until a failure is induced. For instance, the sequence *change table color to black → adjust light level to 50% → set table size to X* yields an environment with a black table of size X, and $50\%$ lighting. The reward function assigns a positive scalar when the outcome is a failure, and a negative scalar when the outcome is a success.

### 3.4 FROM PREDICTING VULNERABILITIES TO IMPROVING ROBOT POLICY PERFORMANCE

The agent $\pi_{\text{MD}}$ outputs a probability distribution over variations (actions), which can directly be interpreted as a map of failure likelihoods. This allows us to both identify highly vulnerable conditions and use them to guide manipulation policy improvement. In the embedding space constructed in Section 3.2, $\pi^{\text{MD}}(a \mid s)$ is modeled as a Gaussian density $p(a \mid s)$ on $\mathbb{R}^{512}$. Although $p(a_t) = 0$ for any exact action, likelihood ratios are well defined: $\frac{p^{\text{MD}}(a_{t1}|s)}{p^{\text{MD}}(a_{t2}|s)}$, indicating which variation is more failure-prone, analogous to PPO's probability-ratio objective. The confidence of these likelihood estimates can be found using the proximity of the current state embedding $e_s$ to the nearest $e \in \mathcal{E}$, measured by $\min_{e \in \mathcal{E}} \|e - e_s\|$. When restricted to a finite candidate set $\mathcal{E}_{\text{known}}$, as in the special case described in Section 3.2, the same mechanism reduces to a categorical distribution over $\mathcal{E}_{\text{known}}$, where mass gradually concentrates on failure-inducing variations. In such cases, likelihoods are given by $\pi^{\text{MD}}(a \mid s) = \frac{\exp(f_a(s))}{\sum_{a'} \exp(f_{a'}(s))}$, which is a probability mass function (PMF) over the discrete action set $\mathcal{A}$, where $f_a(s)$ is the logit for action $a$.

**Improving $\pi_R$ with findings from $\pi_{MD}$.** The likelihood of actions yields a prioritized list of failure modes, which allows practitioners to move beyond collecting broad, unfocused rollouts. Instead, we can target data collection on the highest likelihood failures (e.g., specific lighting or object variations), and fine-tune $\pi_R$ on this compact dataset to systematically patch vulnerabilities, as we will demonstrate in Section 4.4.

## 3.5 THEORETICAL UNDERPINNINGS AND GUARANTEES

We establish that the potential field we build in embedding space does not negatively affect the overall reward but makes the convergence faster.

**Theorem 1** (**Advantage Invariance in a Semantic Potential Field**). *Let $\pi_{MD}$ be the policy for the MDP defined in Sec 3. Let the continuous action space be structured by the embedding $e$, which is trained via the contrastive loss $\mathcal{L}$ (Eq 1) to function as a potential function, $\Phi(s) = e_s$. The search performed by $\pi_{MD}$ in this space is equivalent to learning in a shaped MDP where the implicit shaping function is $F(s, a, s') = \gamma\Phi(s') - \Phi(s)$, for which the following hold: (i) Optimality: any optimal policy in the shaped MDP is also optimal in the original MDP. (ii) Advantage invariance: $A^*(s, a)_{shaped} = A^*(s, a)_{original}$, indicating that the relative advantage is invariant.*

*Proof Sketch.* From potential-based shaping theory Ng et al. (1999), $Q^*_{\text{shaped}}(s, a) = Q^*_{\text{orig}}(s, a) - \Phi(s)$ and $V^*_{\text{shaped}}(s) = V^*_{\text{orig}}(s) - \Phi(s)$, where $Q$ and $V$ are $Q$ and value functions, respectively. Subtracting shows the $\Phi(s)$ cancels, proving advantage invariance. (Full proof in Appendix A)  □

Having established the correctness of the reward, we now demonstrate that it facilitates efficient exploration of the failure–success boundary while enabling faster convergence.

**Theorem 2** (**Sample-Efficient Boundary Exploration**). *Let actions $a \in \mathcal{A}$ be mapped by an $L$-Lipschitz embedding $e(a)$. If $\pi_{MD}$ is trained with reward $R(a) = R_{task}(a) + \beta H(a)$ for $\beta > 0$, where $H(a)$ is predictive uncertainty, then: (i) $R(a)$ is Lipschitz, yielding stable gradients for PPO. (ii) Exploration concentrates near the success/failure boundary, identifying it to precision $\epsilon$ with rollouts polynomial in $1/\epsilon$.*

*Proof Sketch.* The semantic potential induced by the multimodal embedding $e(s)$, trained with BCE and contrastive loss is Lipschitz continuous (proof in Lemma 1 in Appendix A), which ensures smooth rewards and stable gradients. The information-gain drives exploration toward uncertain boundary regions, yielding sample-efficient discovery. (Full proof in Appendix A)  □

**Theorem 3** (**Convergence Acceleration Due to Potential Field**). *Let PPO train a critic with Bellman updates of the form $\|\xi_{t+1}\|_\infty \leq \gamma\|\xi_t\|_\infty + \epsilon$ with critic error $e$ and approximation error $\epsilon$. If potential shaping induces a transformed critic with smaller initialization error $\xi'_0 < \xi_0$ and smaller approximation error $\epsilon' < \epsilon$, then for any $\varepsilon > \epsilon'/(1 - \gamma)$ with discount factor $\gamma$, the shaped critic reaches $\|\xi'_T\|_\infty \leq \varepsilon$ in fewer iterations than the unshaped critic.*

*Proof Sketch.* Due to Theorem 1, the error recursion solves to $\|\xi_T\|_\infty \leq \gamma^K e_0 + \frac{1-\gamma^T}{1-\gamma}\epsilon$. Since both $\xi'_0$ and $\epsilon'$ are strictly smaller, the shaped process crosses any target $\varepsilon$ earlier, implying faster convergence of PPO. (Full proof in Appendix A)  □

## 4 EXPERIMENTAL RESULTS

In this section, we present a series of experiments designed to validate RoboMD. Our evaluation is structured around three central research questions:

1. How does RoboMD's failure diagnosis compare to alternative approaches, including other RL algorithms and VLMs?
2. How effectively does RoboMD generalize its diagnostic capabilities from a known set of perturbations to entirely unseen environmental variations?
3. Do the failure modes identified by RoboMD provide actionable insights that lead to measurable improvements in the robustness of a pre-trained policy?

## 4.1 BENCHMARK COMPARISONS

*Experimental setup*: Our simulated experiments are conducted in RoboSuite Zhu et al. (2020) using datasets from RoboMimic Mandlekar et al. (2021) and MimicGen Mandlekar et al. (2023) (Fig. 7). We evaluate across four standard tasks: lift, stack, threading, and pick & place, which represent a range of manipulation challenges with varying levels of difficulty. These tasks are tested against a diverse set of standard manipulation policies, including Behavior Cloning (BC), Hierarchical BC (HBC), BC-Transformer, Batch Constrained Q-Learning (BCQ), and Diffusion policies.

We first benchmark $\pi_{\text{MD}}$ against a suite of baselines to answer our *first research question* regarding comparative performance. To this end, we construct a dataset of 500 success-failure pairs,

| Reinforcement Learning Models | | | | |
|---|---|---|---|---|
| Model | Lift | Square | Pick Place | Avg. Score |
| A2C | 74.2% | 79.0% | 72.0% | 75.0 |
| PPO | **82.3%** | **84.0%** | **76.0%** | **80.7** |
| SAC | 51.2% | 54.6% | 50.8% | 52.2 |
| Vision–Language Models | | | | |
| Qwen2-VL | 32.0% | 24.6% | **57.4%** | 38.0 |
| Gemini 1.5 Pro | **59.0%** | 36.4% | 37.4% | 44.3 |
| GPT-4o | 57.0% | 44.0% | 32.0% | 33.3 |
| GPT-4o-ICL (5 Shot) | 57.4% | **48.6%** | 57.0% | **54.3** |
| Small Models (Appendix B) | | | | |
| CNN | 46.0% | **57.0%** | 49.0% | **50.6** |
| ResNet | 49.0% | 52.0% | 44.0% | 48.3 |

Table 1: Benchmark results (Accuracy) comparing RL controllers, VLM's, and lightweight models each paired with a BC-MLP low-level policy. $\pi_{MD}$ consistently outperforms other baselines.

| Algorithm | Lift | Pick Place | Threading | Stack |
|---|---|---|---|---|
| BC | 82.5% | 76.0% | 68.0% | 88.0% |
| BCQ | 61.5% | 72.5% | 62.0% | 72.0% |
| HBC | 83.5% | 79.0% | 73.0% | 81.0% |
| BC Transformer | 74.5% | 72.0% | 82.0% | 70.5% |
| Diffusion | 85.0% | 71.0% | 71.0% | 62.0% |

Table 2: RoboMD failure detection accuracy across different tasks.

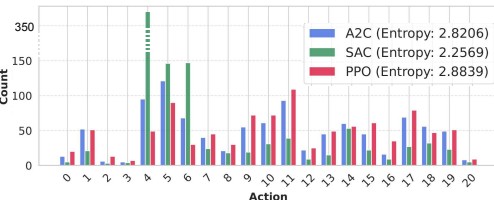

Figure 5: Action diversity across algorithms.

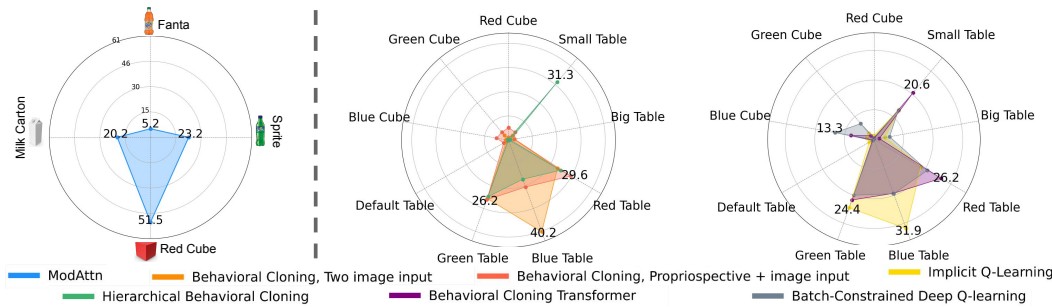

Figure 6: Diagnosis of manipulation policies. Each plot shows failure likelihood ($\propto$ radius) for different actions across environments (real-world in left, and simulation middle and right).

where each pair consists of a randomly selected success and failure case. Since a successful action will rank higher than a failure, this provides ground truth to evaluate $\pi_{MD}$'s ranking consistency. The results, summarized in Table 1, demonstrate that $\pi_{MD}$ outperforms all other methods in ranking accuracy across all tasks. We also conducted evaluations with state-of-the-art proprietary models (GPT-4o and Gemini 1.5 Pro) and an open-source model (Qwen2-VL). Additionally, we extended the evaluation of GPT-4o by employing in-context learning (ICL) with 5-shot demonstrations to gauge its adaptability, ICL improves the performance of GPT-4o, particularly in the *Square* task. However, overall VLM performance remains below 60%, indicating that these models struggle with reliably predicting environment configurations. To compare exploration across RL algorithms (i.e., why PPO?), we analyze the action distributions of A2C Mnih (2016), SAC Haarnoja et al. (2018), and PPO over 21 environment variations (See Appendix F.2). As shown in Fig. 5 and Table 4, PPO achieves the highest entropy (2.8839), indicating its suitability for broader exploration needed for failure discovery. We further evaluate $\pi_{MD}$'s failure detection performance in a variety of standard policies trained using different training methods. The results in Table 2 demonstrate that $\pi_{MD}$ generalizes well across different tasks and policy architectures.

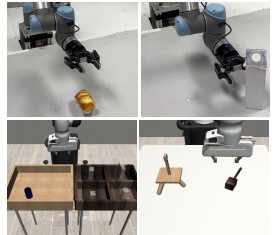

Figure 7: (Top) real world, (Bottom) simulation. Refer Appendix C for all variations.

## 4.2 ANALYZING DIAGNOSTIC CAPABILITIES

We now evaluate $\pi_{MD}$'s ability to diagnose unseen and seen environments, addressing our *second research question*.

**Unseen Environments**: The primary strength of $\pi_{MD}$ is its ability to generalize. We test this by asking $\pi_{MD}$ to rank the failure likelihood of variations it has never seen during training. The results in Table 3 shows in both simulation and the real world, $\pi_{MD}$ correctly ranks the failure likelihood of unseen items (e.g., the "Bread" for the UR5e, "Black Table" in simulation) relative to known items. The consistency of these rankings validates that $\mathcal{E}$ successfully captures the semantic relationships

Table 3: In real ($a_r$: {*Bread* [unseen], *Red Cube*, *Milk Carton*, *Sprite*}) vs. simulated ($a_s$: {*Red Table*, *Black Table* [unseen], *Green Lighting*}) environments, rank consistency measures ordering agreement, and accuracy is computed over 21 unseen variations.

| Task ID | Algorithm | Continuous Rank | Ground Truth Rank | Consistency | Accuracy ($\uparrow$) |
|---|---|---|---|---|---|
| Real Robot (UR5e) | ModAttn Zhou et al. (2022) | $a_r1 > a_r2 > a_r3 > a_r4$ | $a_r1 > a_r2 > a_r3 > a_r4$ | ✓ | - |
| Sim. Can | HBC Mandlekar et al. (2020) | $a_s1 > a_s2 > a_s3$ | $a_s1 = a_s2 > a_s3$ | ✓ | 61% |
| Sim. Square | Diffusion Chi et al. (2023) | $a_s1 > a_s2 > a_s3$ | $a_s1 = a_s2 > a_s3$ | ✓ | 68% |
| Sim. Stack | BCQ Fujimoto et al. (2019) | $a_s1 > a_s2 > a_s3$ | $a_s1 = a_s2 > a_s3$ | ✓ | 80% |
| Sim. Threading | BC Transformer | $a_s1 > a_s2 > a_s3$ | $a_s1 = a_s2 > a_s3$ | ✓ | 74% |

Table 4: Failure characteristics for known action spaces. FSI indicate robustness.

| Model | Entropy ($\downarrow$) | NFM ($\downarrow$) | FSI ($\downarrow$) |
|---|---|---|---|
| IQL Kostrikov et al. (2021) | 2.49 | 4 | 0.72 |
| BCQ Fujimoto et al. (2019) | 2.79 | 6 | 1.15 |
| BC Transformer | 2.47 | 5 | 0.98 |
| HBC Mandlekar et al. (2020) | 2.11 | 4 | 0.68 |
| BC (Two-Image Input) | 2.14 | 3 | 0.63 |
| BC (Proprioceptive + Image) | 2.58 | 3 | 0.75 |
| SmolVLA Shukor et al. (2025) | 1.76 | 2 | 0.73 |
| $\pi0$ Black et al. (2024) | 1.40 | 1 | 0.77 |
| GR00T Bjorck et al. (2025) | 1.54 | 2 | 0.82 |

Table 5: $\pi_R$ fine-tuning strategies. Errors are w.r.t. an ideal policy. Refer Fig. 9.

| Fine-tuning (FT) Strategies | Accuracy % ($\uparrow$) | Mean Square Error ($\downarrow$) | Chi-Square Error ($\downarrow$) |
|---|---|---|---|
| Pre-trained | 67.91 | 0.377 | 0.016 |
| FT with RoboMD | **92.83** | **0.033** | **0.001** |
| FT with 1 failures | 71.25 | 0.068 | 0.002 |
| FT with 2 failures | 75.83 | 0.140 | 0.006 |
| FT with 3 failures | 75.41 | 0.050 | 0.002 |
| FT with 4 failures | 80.00 | 0.128 | 0.005 |
| FT with all failures | 85.48 | 0.069 | 0.003 |

that govern policy failure. As **ablations**, we also test the quality of the $\mathcal{E}_{known}$ by comparing three configurations: using only BCE loss, using BCE with and our contrastive loss, and an image-only backbone model. The results are shown in Fig. 8 and Table 9. The confusion matrix for the full model (Image+Text with BCE+Contrastive loss) shows a strong diagonal structure, indicating high separability between different actions. Quantitatively, this model achieves the lowest MSE (0.1801) and Frobenius distance (7.6387) from an ideal identity matrix. This confirms that multimodal inputs and a contrastive loss is crucial for creating a locally consistent embedding space.

**Seen Environments**: Fig. 6 visualizes the failure likelihoods that $\pi_{MD}$ generates for different manipulation policies. HBC policy shows high robustness, which is quantitatively confirmed by its low Failure Severity Index (FSI) and small number of failure modes (NFM) in Table 4. FSI quantifies the weighted impact defined by $\sum_{i=1}^{N} P_{failure}(a_i) \cdot W_i$ where $P_{failure}$ represents the probability of failure for action $a_i$, and $W_i$ is the normalized weight such that the failure with the highest probability is assigned a weight of 1. The significant variation in failure patterns across different policies (e.g., BC Transformer vs. HBC) is expected, as different architectures have unique inductive biases and thus different weaknesses, all of which are effectively captured by $\pi_{MD}$.

## 4.3 ABLATION: EMBEDDING QUALITY AND SEMANTIC CONTINUITY

RoboMD relies on a vision–language embedding that must both separate success and failure modes and vary smoothly under physical changes. As shown in Table 6, the learned embedding forms well-separated and compact clusters across simulated tasks and real-world SO101 policies, with large separation ratios, high effect sizes, strong silhouette scores, and low Davies–Bouldin indices. In addition, Table 7 demonstrates semantic continuity under a controlled lighting-intensity sweep for a BC policy trained to pick up a cube under normal lighting, where cumulative embedding distance

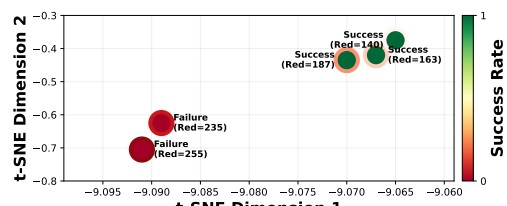

Figure 10: Embedding space $\mathcal{E}$ demonstrates a clear semantic continuity between failure and success states: as the red light intensity is decreased, the model transitions smoothly from failure clusters to success clusters.

increases monotonically as illumination decreases (Kendall's $\tau = 1.000$, Pearson's $r = 0.982$). Together, these results confirm that the embedding is both structurally well-organized and semantically smooth, justifying its use as a continuous potential field for PPO-based exploration.

## 4.4 ACTIONABLE INSIGHTS: FAILURE-GUIDED $\pi_R$ POLICY IMPROVEMENT

Finally, we address our *third research question*: whether the diagnostic signal provided by $\pi_{MD}$ can be directly leveraged to improve policy robustness. Using the top-ranked failures identified by $\pi_{MD}$, we construct a targeted dataset for fine-tuning a pre-trained BC-lift manipulation policy. As shown in Fig. 9, vulnerabilities exhibited by the pre-trained policy are substantially mitigated after RoboMD-

Table 6: Embedding-space grouping metrics across simulated tasks and real-world policies. High separation ratio (Sep.), silhouette score (Silh.), together with a low Davies-Bouldin (DB) index.

| Task | Sep. ($\uparrow$) | Silh. ($\uparrow$) | DB ($\downarrow$) |
|---|---|---|---|
| Lift – BC | 22.99 | 0.91 | 0.15 |
| Square – BC | 20.23 | 0.88 | 0.25 |
| SO101 – SmolVLA | 10.37 | 0.87 | 0.35 |
| SO101 – ACT | 7.76 | 0.84 | 0.38 |
| SO101 – $\pi_0$ | 23.97 | 0.95 | 0.07 |

Table 7: Monotonic embedding-distance progression under decreasing lighting intensity. Cumulative distance from the initial demonstration increases smoothly as illumination is reduced, demonstrating semantic continuity.

| Demo | Success | C. Dist. | Step Dist. |
|---|---|---|---|
| Demo 1 (High) | 0 | 0.00 | 0.00 |
| Demo 2 | 0 | 0.22 | 0.22 |
| Demo 3 | 1 | 0.55 | 0.35 |
| Demo 4 | 1 | 0.58 | 0.07 |
| Demo 5 (Low) | 1 | 0.67 | 0.11 |



Figure 8: Similarity matrix of embeddings trained using a) BCE, b) BCE+Contrastive loss, and c) no text encoder.

Table 9: Measuring embedding quality across different loss functions. Lower MSE and Frobenius-norm distances (to the identity) indicate embeddings closer to the ideal diagonal (better separation).

| Eval Loss | Image BCE | Image BCE+Contr. | Image+Text BCE | Image+Text BCE+Contr. |
|---|---|---|---|---|
| MSE ($\downarrow$) | 0.6495 | 0.6179 | 0.8426 | 0.1801 |
| Fro dist. ($\downarrow$) | 14.5060 | 14.1497 | 16.5227 | 7.6387 |

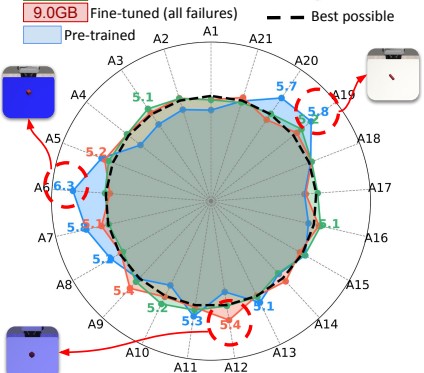

Figure 9: Failure distribution before and after fine-tuning. Observe reduction in error modes. Dashed black line represents no failure. Refer Table 5.

guided fine-tuning. Beyond qualitative improvements, Table 5 quantitatively demonstrates that fine-tuning with RoboMD-selected failures achieves a significant accuracy gain of **92.83%**, outperforming fine-tuning on randomly selected failures (**85.48%**) while using substantially less data.

Importantly, RoboMD-guided fine-tuning requires only **1.3 GB** of targeted failure data, compared to **9.0 GB** when fine-tuning on all known failures. This highlights that $\pi_{\text{MD}}$ does not merely expose failure modes, but also prioritizes the most informative failures for policy adaptation. Consequently, RoboMD enables efficient, targeted fine-tuning that improves robustness under constrained data collection budgets.

## 5 CONCLUSION

This paper introduced *RoboMD*, a framework that recasts the diagnosis of robot manipulation failures from a process of manual heuristics into an active, sequential search problem solved by deep reinforcement learning. By training a diagnostic agent to systematically seek out failure-inducing conditions, RoboMD provides a principled approach to uncovering policy weaknesses. Our key contribution, lifting this search into a continuous vision-language embedding space enables the RL agent to generalize its diagnosis, successfully identifying and ranking subtle failure modes in both seen and previously unseen environmental conditions. Our extensive experiments validate this methodology, demonstrating that RoboMD significantly outperforms static VLM baselines, small models and alternative RL methods in diagnostic accuracy. We also showed that the identified failures provide an actionable path to enhancing policy robustness, allowing for targeted fine-tuning that measurably improves performance in high-risk scenarios at a fraction of data cost. Beyond debugging, RoboMD offers a practical foundation for integration into continuous testing pipelines, enabling automated regression testing as policies evolve. Future work will focus on scaling this approach to create generalist diagnostic agents capable of evaluating multi-task policies across even broader and more unstructured environment distributions, a crucial step toward the deployment of truly reliable and trustworthy robotic systems in the real world.

## 6 ETHICS STATEMENT

This work adheres to the ICLR Code of Ethics. Our research does not involve human subjects, personally identifiable data, or sensitive information. All datasets used are publicly available and licensed for research purposes. We have made sure that our methods and results are reported honestly, transparently, and reproducibly. The potential societal impacts of this work were considered, and our contributions align very well with responsible and beneficial use of safe machine learning model deployment.

## 7 REPRODUCIBILITY STATEMENT

We have taken several measures to ensure the reproducibility of our work. An anonymous GitHub repository is provided that contains all the necessary scripts to reproduce the experiments and results. The simulation data used in this study is publicly available. For real-world experiments, we employ a commonly used UR5 robotic arm setup, whose specifications and control interface are well documented. Detailed experimental setup is provided in the Appendix C. Together, these resources allow independent researchers to fully replicate our results.

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

APPENDIX

## A  PROOFS OF THEORETICAL GUARANTEES

**Theorem 1** (**Advantage Invariance in a Semantic Potential Field**). *Let $\pi^{MD}$ be the policy for the MDP defined in Section 3. Let the continuous action space be structured by the embedding $e$, which is trained via the contrastive loss $\mathcal{L}$ (Eq. 2) to function as a potential function, $\Phi(s) = e_s$. The search performed by $\pi^{MD}$ in this space is equivalent to learning in a shaped MDP where the implicit shaping function is $F(s, a, s') = \gamma\Phi(s') - \Phi(s)$. For this process, the following hold:*

*(i) **Optimality:** Any optimal policy in the shaped MDP is also optimal in the original MDP.*

*(ii) **Advantage Invariance:** $A^*(s, a)_{shaped} = A^*(s, a)_{original}$.*

*Proof.* The proof relies on the established theory of potential-based reward shaping Ng et al. (1999). Let the original MDP be $M$ with reward function $R(s, a, s')$ and the shaped MDP be $M'$ with reward function $R'(s, a, s') = R(s, a, s') + F(s, a, s')$, where $F(s, a, s') = \gamma\Phi(s') - \Phi(s)$ is the potential-based shaping reward.

The optimal Q-function in the original MDP, $Q_M^*$, satisfies the Bellman optimality equation:

$$Q_M^*(s, a) = \mathbb{E}_{s' \sim P_{sa}(\cdot)}[R(s, a, s') + \gamma \max_{a'} Q_M^*(s', a')]$$

Now, consider a transformed Q-function, $\hat{Q}(s, a) = Q_M^*(s, a) - \Phi(s)$. We can show that it satisfies the Bellman equation for the shaped MDP, $M'$:

$$\hat{Q}(s, a) + \Phi(s) = \mathbb{E}_{s' \sim P_{sa}(\cdot)}[R(s, a, s') + \gamma \max_{a'}(\hat{Q}(s', a') + \Phi(s'))]$$

$$\hat{Q}(s, a) = \mathbb{E}_{s' \sim P_{sa}(\cdot)}[R(s, a, s') - \Phi(s) + \gamma\Phi(s') + \gamma \max_{a'} \hat{Q}(s', a')]$$

$$= \mathbb{E}_{s' \sim P_{sa}(\cdot)}[R(s, a, s') + F(s, a, s') + \gamma \max_{a'} \hat{Q}(s', a')]$$

$$= \mathbb{E}_{s' \sim P_{sa}(\cdot)}[R'(s, a, s') + \gamma \max_{a'} \hat{Q}(s', a')]$$

This is precisely the Bellman optimality equation for $M'$. By the uniqueness of the optimal Q-function, it must be that $Q_{M'}^*(s, a) = \hat{Q}(s, a) = Q_M^*(s, a) - \Phi(s)$.

**(i) Proof of Optimality:** An optimal policy $\pi^*$ is one that acts greedily with respect to the optimal Q-function. For the shaped MDP, the optimal policy is:

$$\pi_{M'}^*(s) = \arg\max_a Q_{M'}^*(s, a)$$
$$= \arg\max_a (Q_M^*(s, a) - \Phi(s))$$

Since $\Phi(s)$ does not depend on the action $a$, it does not change the argmax. Therefore:

$$\pi_{M'}^*(s) = \arg\max_a Q_M^*(s, a) = \pi_M^*(s)$$

Thus, any optimal policy for the shaped MDP is also optimal for the original MDP.

**(ii) Proof of Advantage Invariance:** The advantage function is defined as $A(s, a) = Q(s, a) - V(s)$, where $V(s) = \max_a Q(s, a)$. Using the relationship derived above:

$$V_{M'}^*(s) = \max_a Q_{M'}^*(s, a) = \max_a(Q_M^*(s, a) - \Phi(s)) = (\max_a Q_M^*(s, a)) - \Phi(s) = V_M^*(s) - \Phi(s)$$

Now, we can write the advantage function for the shaped MDP:

$$A_{M'}^*(s, a) = Q_{M'}^*(s, a) - V_{M'}^*(s)$$
$$= (Q_M^*(s, a) - \Phi(s)) - (V_M^*(s) - \Phi(s))$$
$$= Q_M^*(s, a) - V_M^*(s) = A_M^*(s, a)$$

This proves that the advantage function is invariant under potential-based reward shaping. $\square$

**Theorem 2** (**Sample-Efficient Boundary Exploration**). *Let actions $a \in \mathcal{A}$ be mapped by an L-Lipschitz embedding $\mathbf{e}(a)$. If $\pi^{MD}$ is trained with reward $R(a) = R_{task}(a) + \beta H(a)$ for $\beta > 0$, where $H(a)$ is predictive uncertainty, then: (i) $R(a)$ is Lipschitz, yielding stable gradients for PPO; (ii) exploration concentrates near the success/failure boundary, identifying it to precision $\epsilon$ with rollouts polynomial in $1/\epsilon$.*

**Lemma 1** (Lipschitzness of the shaped reward). *Let $\mathbf{e} : \mathcal{A} \to \mathcal{E}$ be $L_{\mathbf{e}}$-Lipschitz. Assume $R_{task} : \mathcal{E} \to \mathbb{R}$ and $H : \mathcal{E} \to \mathbb{R}$ are $L_{R_{task}}$- and $L_H$-Lipschitz, respectively. For $\beta > 0$, define $R(a) = R_{task}(\mathbf{e}(a)) + \beta H(\mathbf{e}(a))$. Then $R$ is Lipschitz on $\mathcal{A}$ with constant*

$$L_R \leq L_{\mathbf{e}}\big(L_{R_{task}} + \beta L_H\big).$$

*Proof.* For any $a_1, a_2 \in \mathcal{A}$,

$$
\begin{aligned}
|R(a_1) - R(a_2)| &= \big|R_{\text{task}}(\mathbf{e}(a_1)) - R_{\text{task}}(\mathbf{e}(a_2)) + \beta\big(H(\mathbf{e}(a_1)) - H(\mathbf{e}(a_2))\big)\big| \\
&\leq |R_{\text{task}}(\mathbf{e}(a_1)) - R_{\text{task}}(\mathbf{e}(a_2))| + \beta\,|H(\mathbf{e}(a_1)) - H(\mathbf{e}(a_2))| \\
&\leq L_{R_{\text{task}}}\,\|\mathbf{e}(a_1) - \mathbf{e}(a_2)\| + \beta L_H\,\|\mathbf{e}(a_1) - \mathbf{e}(a_2)\| \\
&\leq \big(L_{R_{\text{task}}} + \beta L_H\big)L_{\mathbf{e}}\,\|a_1 - a_2\|.
\end{aligned}
$$

Thus $R$ is Lipschitz with the stated constant. $\qquad\square$

**Standing assumptions.** Let $p(a) = \Pr[\text{failure} \mid a]$ denote the classifier's predicted probability. Assume:

A1 (*Calibration + smoothness*) $p$ is calibrated and $L_p$-Lipschitz in $\mathbf{e}(a)$.

A2 (*Regular boundary*) There exist $r_0 > 0$ and $\kappa > 0$ such that for all $a$ with $\mathrm{dist}(a, \partial\mathcal{S}) \leq r_0$, $\big|p(a) - \frac{1}{2}\big| \geq \kappa\,\mathrm{dist}(a, \partial\mathcal{S})$, where $\partial\mathcal{S} = \{a : p(a) = \frac{1}{2}\}$.

**Lemma 2** (Uncertainty maximization at/near the boundary). *Let $H(a) = -p(a)\log p(a) - (1 - p(a))\log(1 - p(a))$ be the predictive entropy. Under (A1)–(A2), there exist constants $c_H > 0$ and $r_0 > 0$ such that for all $a$ with $\mathrm{dist}(a, \partial\mathcal{S}) \leq r_0$,*

$$H(a) \leq \log 2 - c_H\big(p(a) - \tfrac{1}{2}\big)^2 \leq \log 2 - c_H\,\kappa^2\,\mathrm{dist}(a, \partial\mathcal{S})^2,$$

*and consequently $H$ achieves its maxima on $\partial\mathcal{S}$ and decays quadratically away from it in distance.*

*Proof.* The binary entropy $h(p) = -p\log p - (1 - p)\log(1 - p)$ is strictly concave. It attains its maximum at $p = \frac{1}{2}$ with value $h(1/2) = \log 2$, and satisfies $h'(1/2) = 0$ and $h''(1/2) = -4$. By Taylor's theorem, for $p$ in a neighborhood of $\frac{1}{2}$ there exists a constant $c_H \in (0, 2]$ such that

$$h(p) \leq \log 2 - c_H\,(p - \tfrac{1}{2})^2.$$

Applying this with $p = p(a)$ gives

$$H(a) \leq \log 2 - c_H\,(p(a) - \tfrac{1}{2})^2.$$

Moreover, Assumption (A2) states that whenever $\mathrm{dist}(a, \partial\mathcal{S}) \leq r_0$,

$$\big|p(a) - \tfrac{1}{2}\big| \geq \kappa\,\mathrm{dist}(a, \partial\mathcal{S}).$$

Combining the two inequalities yields

$$H(a) \leq \log 2 - c_H\kappa^2\,\mathrm{dist}(a, \partial\mathcal{S})^2.$$

Thus $H(a)$ is maximized on the boundary $\partial\mathcal{S}$ and decays quadratically with distance away from it. $\qquad\square$

**Lemma 3** (Sample complexity of boundary identification). *Suppose $f(a) := p(a) - \frac{1}{2}$ admits a Gaussian process (GP) surrogate with kernel $k$, and let $\gamma_T$ denote the maximal information gain after $T$ queries for $k$. Consider active sampling driven by the shaped reward $R = R_{task} + \beta H$ with $\beta > 0$. Then there exists a constant $C > 0$ such that, with high probability, the zero level set $\{a : f(a) = 0\}$ (i.e., the boundary) is identified to Hausdorff precision $\epsilon$ after at most*

$$T \leq C\,\frac{\gamma_T \log T}{\epsilon^2}$$

*queries.*

*Proof sketch.* Under (A1)–(A2), Lemma 2 shows that, in a neighborhood of the boundary, entropy $H$ is a smooth, strictly decreasing function of $|f(a)|$ and is maximized where $f(a) = 0$. Hence maximizing $\beta H$ is equivalent (up to smooth monotone reparameterization) to prioritizing high posterior uncertainty on the sign of $f$ near its zero level set, as in standard level-set acquisitions (e.g., straddle/variance/ambiguity criteria).

For a GP posterior, uniform confidence bands $|f(a) - \mu_{T-1}(a)| \le \alpha_T^{1/2}\sigma_{T-1}(a)$ hold with high probability. Sampling high-variance points in the ambiguous band around $f = 0$ shrinks that band until its thickness is $O(\epsilon)$. The cumulative posterior variance is controlled by the information gain $\gamma_T$, which yields

$$T = O\Big(\frac{\gamma_T \log T}{\epsilon^2}\Big)$$

for $\epsilon$-accurate recovery of the zero level set; see, e.g., Theorem 1 of Gotovos et al. (2013). Finally, since $H$ and these ambiguity/variance scores agree up to a smooth monotone transform near the boundary (Lemma 2), the same rate applies to the $\beta H$-driven policy. $\square$

*Proof of Theorem 2.* Part (i): By Lemma 1, $R$ is Lipschitz with constant $L_R \le L_{\mathbf{e}}(L_{R_{\text{task}}} + \beta L_H)$, which yields bounded PPO gradients.

Part (ii): By Lemma 2, $H$ is maximized on (and decays away from) the decision boundary, so the $\beta H$ term concentrates exploration in an $O(\epsilon)$ tube around it. Lemma 3 then gives the $T = O\big(\frac{\gamma_T \log T}{\epsilon^2}\big)$ rollout bound to achieve precision $\epsilon$. $\square$

**Theorem 3** (**Convergence Acceleration Due to Potential Field**). *Let PPO train a critic with Bellman updates of the form $\|\xi_{t+1}\|_\infty \le \gamma\|\xi_t\|_\infty + \epsilon$ with critic error $e$ and approximation error $\epsilon$. If potential shaping induces a transformed critic with smaller initialization error $\xi_0' < \xi_0$ and smaller approximation error $\epsilon' < \epsilon$, then for any $\varepsilon > \epsilon'/(1-\gamma)$ with discount factor $\gamma$, the shaped critic reaches $\|\xi_T'\|_\infty \le \varepsilon$ in fewer iterations than the unshaped critic.*

*Proof.* Unroll the linear recursions. For any $t \ge 0$,

$$\|\xi_t\|_\infty \ \le \ \gamma^t\xi_0 \ + \ \frac{1-\gamma^t}{1-\gamma}\,\epsilon \ = \ \frac{\epsilon}{1-\gamma} \ + \ \gamma^t\Big(\xi_0 - \frac{\epsilon}{1-\gamma}\Big), \tag{1}$$

and similarly

$$\|\xi_t'\|_\infty \ \le \ \gamma^t\xi_0' \ + \ \frac{1-\gamma^t}{1-\gamma}\,\epsilon' \ = \ \frac{\epsilon'}{1-\gamma} \ + \ \gamma^K\Big(\xi_0' - \frac{\epsilon'}{1-\gamma}\Big). \tag{2}$$

Define the minimal hitting times for a target tolerance $\varepsilon > 0$:

$$\tau \ := \ \min\{t \in \mathbb{N} : \ \|\xi_t\|_\infty \le \varepsilon\}, \qquad \tau' \ := \ \min\{t \in \mathbb{N} : \ \|\xi_t'\|_\infty \le \varepsilon\}.$$

From $e_0' < e_0$ and $\epsilon' < \epsilon$ and the positivity of the coefficients $\gamma^t$ and $\frac{1-\gamma^t}{1-\gamma}$ for $t \ge 0$, we have the pointwise strict inequality

$$\gamma^t\xi_0' + \frac{1-\gamma^t}{1-\gamma}\,\epsilon' \ < \ \gamma^t\xi_0 + \frac{1-\gamma^t}{1-\gamma}\,\epsilon \quad \text{for all } t \ge 0, \tag{3}$$

i.e., the shaped upper bound is strictly below the unshaped one at every $t$.

Now fix any $\varepsilon > \epsilon'/(1-\gamma)$. By (2), there exists a finite $t$ with $\|\xi_t'\|_\infty \le \varepsilon$, so $\tau'$ is finite. If the unshaped process is also able to reach $\varepsilon$ (i.e., $\tau < \infty$), then from (1)–(3) we get

$$\|\xi_\tau'\|_\infty \ \le \ \gamma^\tau\xi_0' + \frac{1-\gamma^\tau}{1-\gamma}\,\epsilon' \ < \ \gamma^\tau\xi_0 + \frac{1-\gamma^\tau}{1-\gamma}\,\epsilon \ \le \ \varepsilon,$$

which implies $\tau' \le \tau$.

Moreover, strict inequality of the shaped bound (3) together with minimality of $\tau$ implies that either $\tau' = \tau - 1$ (or smaller) or, in the degenerate case that $\varepsilon \ge \|\xi_0\|_\infty$, both processes already satisfy the target at $t = 0$. Hence, whenever the unshaped process requires at least one update to reach $\varepsilon$ (i.e., $\tau \ge 1$), we have $\tau' < \tau$. $\square$

Table 10: CNN Confusion matrix percentages (%)

| Task | TP | FN | TN | FP |
|------|-----|-----|-----|-----|
| Square | 39 | 61 | 74 | 26 |
| Can | 7 | 93 | 92 | 8 |
| Lift | 8 | 92 | 87 | 13 |

Table 11: ResNet-18 Confusion matrix percentages (%)

| Task | TP | FN | TN | FP |
|------|-----|-----|-----|-----|
| Square | 60 | 40 | 34 | 66 |
| Can | 10 | 90 | 84 | 16 |
| Lift | 47 | 53 | 51 | 49 |

## B DISCUSSION: FAILURES, OOD, AND NEURAL NETWORK BASELINES

Failures arise both in OOD and in-distribution scenarios, so OOD detectors alone cannot serve as comprehensive failure detectors. While OOD methods can detect novel textures or extreme lighting they miss subtle perturbations that actually cause policy failures. We evaluated two lightweight visual detectors a three layer CNN and a halved-channel ResNet-18 on our collected success–failure frames from the square, can and lift tasks. Table 10 and Table 11 report confusion matrix rates: the CNN achieves true-positive rates of 39 %, 7 % and 8 % with false-positive rates of 26 %, 8 % and 13 %; ResNet-18 reaches true-positives of 60 %, 10 % and 47 % with false-positives of 66 %, 16 % and 49 %. These low true-positive and high false-positive rates show that standalone OOD detectors lack the sensitivity and specificity needed for reliable failure detection in robotic tasks.

As a sanity check on representational capacity, we evaluated the same two models on a balanced set of nominal versus failure-mode frames exhibiting minor lighting shifts, small object-size changes and novel cube/table colors (see Table 1). Both models were trained for 200 epochs on 84×84 RGB inputs normalized to [0,1]. Neither exceeded chance accuracy ($\approx$ 50 %) at ranking failures and success on these fine-grained perturbations. This confirms that low-capacity visual encoders cannot disentangle subtle failure cues, motivating our policy-centric, multimodal RoboMD framework, which leverages semantic embeddings to rank and generalize failure modes under targeted perturbations.

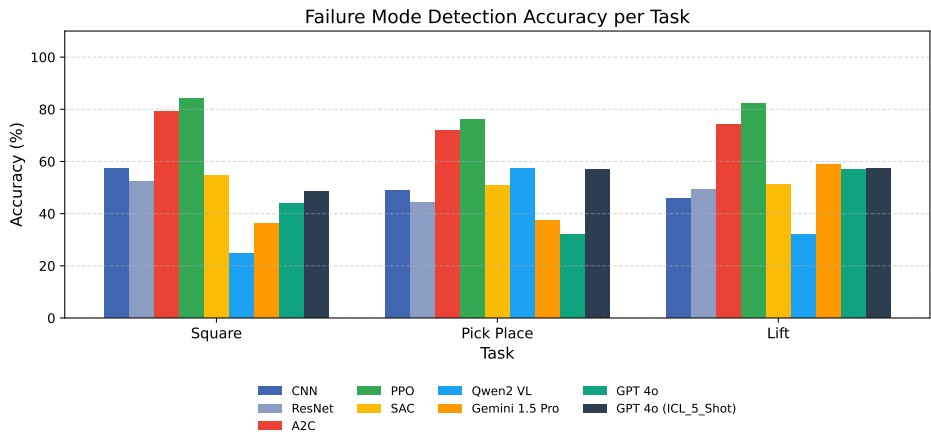

Figure 11: Testing Robustness Under Visual Perturbations: Successful Rollout in Training vs. Failure Induced by Red Table Distraction

The CNN encoder comprised two convolutional layers (3→32→64 channels, 5×5 kernels, stride 2, ReLU), followed by flattening and a 128-dimensional binary classification head. The ResNet-18 variant retained the original residual block design but halved all channel widths (16→32→64→128), applied global average pooling, and used a 64-dimensional classification layer.

Failure detection in robotic systems can be broadly categorized into three approaches:

1. **Out-of-Distribution (OOD) Detection.** These methods model the distribution of nominal training data and flag inputs that deviate from it. They are commonly used for runtime

monitoring in perception and control pipelines Xu et al. (2025). However, not all OOD inputs lead to failures, and many failures occur within the training distribution, limiting their coverage.

2. **Supervised Failure Prediction.** A classifier or regressor is trained on labeled success and failure data to directly predict task outcomes Gu et al. (2025). This approach can capture both in-distribution and OOD failures if represented in the dataset, but its performance depends heavily on the diversity and completeness of observed failure modes.

3. **Active Failure Search and Stress Testing.** Instead of passively predicting failures, these methods actively probe the robot or its environment to discover vulnerabilities. Techniques include adversarial perturbations, domain randomization, and reinforcement learning–based exploration. This paradigm aims to uncover unseen and rare failure modes by systematically searching high-risk regions of the state or environment space.

## C EXPERIMENTAL SETUP

### C.1 COMPUTING RESOURCES

All model training was performed on a single NVIDIA H100 GPU (80 GB HBM2), with peak GPU memory usage of approximately 60 GB. We used mixed-precision (FP16) training under PyTorch 2.0 and CUDA 11.8 to maximize throughput. Full training (one tasks) required roughly 12 hours on this setup.

For inference, the model's footprint falls well below 16 GB of GPU memory, so it can be deployed on a wide range of hardware (e.g., NVIDIA A100 40 GB, RTX 3090) without requiring an H100-class device. All experiments ran on an Ubuntu 22.04 server with 256 GB of system RAM and dual Intel Xeon CPUs.

### C.2 REAL-WORLD EXPERIMENT SETUP

Real-world experiments were conducted using a UR5e robotic arm equipped with high-resolution cameras and a standardized workspace. The setup is shown below in Fig 12.

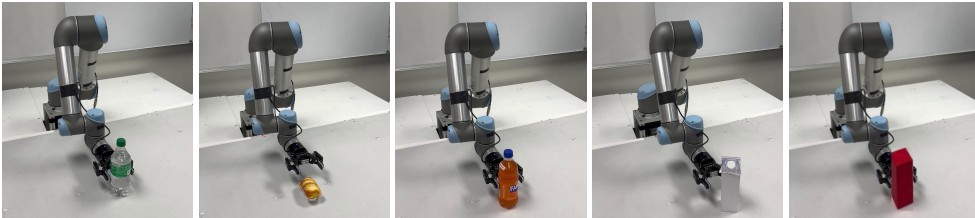

Figure 12: Scenes from experiments on real world robot

### C.3 SIMULATION EXPERIMENT SETUP

Simulation experiments were performed using the MuJoCo physics engine integrated with Robosuite. The simulated environments included variations in object positions, shapes, and textures. The simulation allowed extensive testing across diverse scenarios. Below we show a few samples in Fig 13.

### C.4 SEMANTIC SPACE TRAINING

For image encoding, we adopt the **Vision Transformer Base (ViT-B/16)** architecture from Torchvision, pretrained on the ImageNet-1k dataset (IMAGENET1K_V1 weights). The original classification head is replaced with a linear projection to a 512-dimensional latent space.

For text encoding, we employ the **CLIP text transformer** from OpenAI's **ViT-B/32** model, which provides contextual embeddings of action descriptions. Unlike the vision branch of CLIP, which is

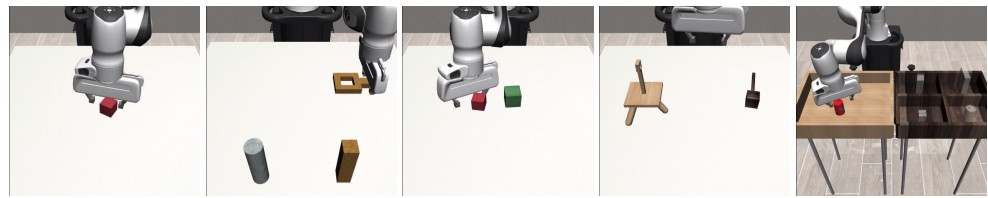

Figure 13: Scenes from experiments on Robosuite

substituted with our ViT-B/16 encoder, the text branch is fine-tuned jointly with the image encoder to learn a shared semantic embedding space.

## D  BASELINES

To validate the effectiveness of our method, we compared it against two categories of baselines: Reinforcement Learning (RL) baselines and Vision-Language Model (VLM) baselines. Below, we detail their implementation, hyperparameters, and specific configurations.

### D.1  REINFORCEMENT LEARNING (RL) BASELINES

The RL baselines were implemented using well-established algorithms, each optimized for the task to ensure a fair comparison. The following RL methods were included:

- **Proximal Policy Optimization (PPO):** A policy-gradient method known for its stability and efficiency. Key hyperparameters included:
    - Learning rate: $3 \times 10^{-4}$
    - Discount factor ($\gamma$): 0.99
    - Clipping parameter ($\epsilon$): 0.2
    - Number of epochs: 10
    - Batch size: 64
    - Actor-Critic network layers: [128, 256, 128]
- **Soft Actor-Critic (SAC):** A model-free off-policy algorithm optimized for continuous action spaces. The key hyperparameters were:
    - Learning rate: $1 \times 10^{-3}$
    - Discount factor ($\gamma$): 0.99
    - Replay buffer size: $1 \times 10^{6}$
    - Target entropy: $-\dim(\text{action space})$
    - Batch size: 128
- **Advantage Actor Critic (A2C):**
    - Learning rate: $2.5 \times 10^{-4}$
    - Discount factor ($\gamma$): 0.99
    - Exploration strategy: Epsilon-greedy ($\epsilon$ decayed from 1.0 to 0.1 over 500,000 steps)
    - Replay buffer size: $1 \times 10^{6}$
    - Batch size: 64
    - Neural network layers: [128, 256, 128]

Each RL baseline was evaluated using the same metrics, ensuring consistency across comparisons.

### D.2  VISION-LANGUAGE MODEL (VLM) BASELINES

The VLM baselines take advantage of the interplay between visual and textual modalities for task representation. We evaluated 3 state-of-the-art VLMs adapted to our task:

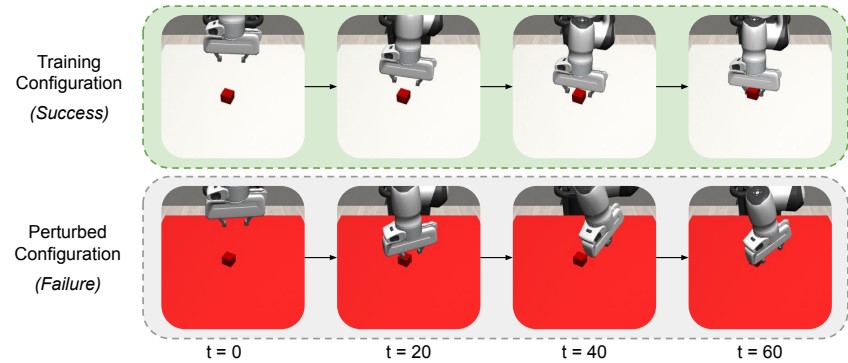

Figure 14: Testing Robustness Under Visual Perturbations: Successful Rollout in Training vs. Failure Induced by Red Table Distraction

1. GPT-4o
2. Gemini 1.5 Pro
3. Qwen2-VL

Additionally, we leverage GPT-4o with in-context learning, using five demonstrations. First, we process the output trajectories into videos and compute the appropriate frame rate to generate video sequences equivalent to 15 frames per trajectory pair. These sequences, representing perturbation scenarios, are provided to the VLMs along with a system prompt that includes a detailed policy description, training configuration, and a natural language task description. For evaluation, we structure the testing dataset using a pairwise comparison framework, where each model is prompted to assess two input video sequences and rank which is more likely to result in task success. The results are recorded in a CSV file, and we compute comparison scores by analyzing model rankings against ground-truth rollouts in the simulated perturbation.

## E  RATIONALE FOR USING REINFORCEMENT LEARNING

RL is employed in the RoboMD framework due to its ability to explore high-dimensional, complex action spaces and optimize sequential decision-making under uncertainty. This section outlines the key motivations for choosing RL as the core methodology:

**Exploration of High-Risk Scenarios**: Traditional approaches to analyzing robot policy failures often rely on deterministic sampling or exhaustive evaluation, which become infeasible in large, dynamic environments. RL allows targeted exploration by learning an agent that actively seeks out environmental configurations likely to induce policy failures. This capability is particularly useful for systematically uncovering vulnerabilities in high-dimensional environments.

**Optimization of Failure Discovery**: The objective of RoboMD is to maximize the occurrence of failures in pre-trained policies. RL frameworks, such as PPO, are well-suited for this task as they iteratively refine policies to achieve specific goals, such as identifying high-risk states. The reward function incentivizes the agent to find configurations where the manipulation policy fails by going through multiple actions to induce failures. Fig 14 shows several steps of the manipulation policy rollout.

**Comparison with Alternative Methods**: While other methods, such as supervised learning or heuristic-based exploration, can provide valuable insights into specific failure cases, they are limited in their scope and adaptability. Supervised learning approaches rely heavily on labeled data, which is challenging to obtain for failure analysis, particularly for rare or unseen failure modes. These methods also lack the ability to adapt dynamically to changes in the environment, reducing their effectiveness in exploring novel or complex failure scenarios. Similarly, heuristic-based exploration methods, such as grid search or predefined sampling strategies, can identify failure cases under controlled conditions but struggle to generalize in high-dimensional environments where the space of possible failure configurations is vast. These methods are also constrained by their reliance on static,

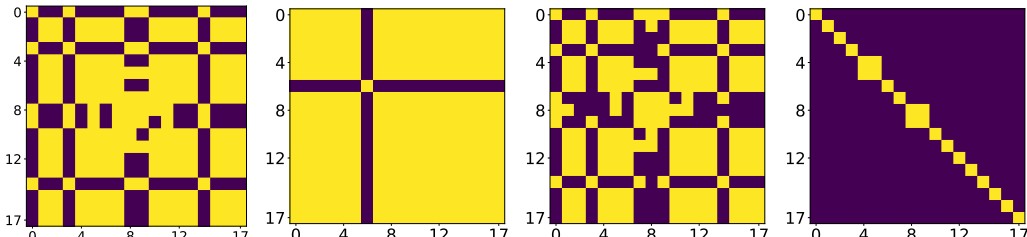

Figure 15: The order in which the confusion matrix is a) Image Ecoder + BCE b) Image + Text Encoder + BCE loss c) Image Encoder + BCE + Contrastive loss d) Image + Text Encoder + BCE + Contrastive loss

predefined rules, which often fail to capture the intricate interactions between environmental factors and failure likelihoods. In contrast, reinforcement learning excels in scenarios where exploration and generalization are critical.

# F    CONTINUOUS ACTION SPACE EMBEDDING

Embedding actions in a continuous space is crucial for efficiently capturing the underlying structure of decision-making processes. Unlike discrete action spaces, where each action is treated as an independent category, continuous action space embeddings aim to encode similarities and relationships between actions in a structured space.

Table 12: Actions for Can and Box tasks.

| Task | Action Description |
|------|--------------------|
| Can | Change the can color to red. |
| | Change the can color to green. |
| | Change the can color to blue. |
| | Change the can color to grey. |
| Box | Change the box color to green. |
| | Change the box color to blue. |
| | Change the box color to red. |
| Box Sizes | Resize the box to $0.3 \times 0.3 \times 0.02$ (L, B, H). |
| | Resize the box to $0.2 \times 0.2 \times 0.02$ (L, B, H). |
| | Resize the box to $0.1 \times 0.1 \times 0.02$ (L, B, H). |

## F.1    ACTION DESCRIPTION MAPPING FOR CLIP LANGUAGE INPUT

To generate language inputs for CLIP, we use a mapped dictionary that encodes the action being applied to the image. The action descriptions for different tasks are detailed in Table 12. This table represents only a subset of possible actions, and users are free to modify the language as needed. The descriptions are not strict requirements, as the model learns over time to associate text and images with failure patterns, allowing for flexibility in phrasing while maintaining the underlying semantic meaning. The actions used for Lift task is as follows which was also shown as (A1,A2...A21) in Fig 9.

## F.2    EVALUATION

Fig. 9 shows the failure distribution in the Lift environment under 21 independent perturbations. These include 4 cube colors, 4 table colors, 3 cube sizes, 2 table sizes, 4 robot colors, and 4 lighting

conditions. Fig 15 illustrates the similarity structure of embeddings trained using only Binary Cross-Entropy (BCE) loss, resulting in highly correlated representations. In contrast, the right matrix, trained with a combination of BCE and Contrastive Loss, demonstrates improved separation, as evidenced by the stronger diagonal structure and reduced off-diagonal similarities.

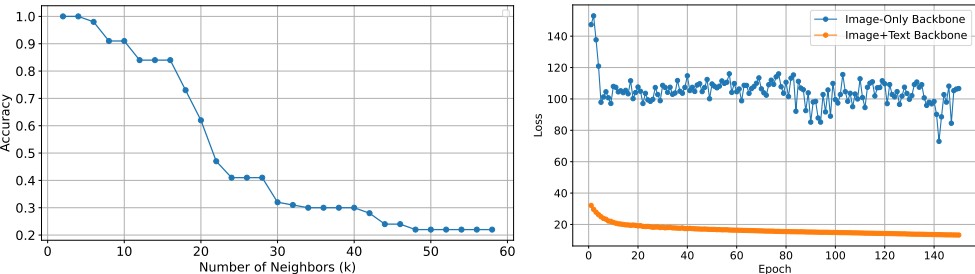

Figure 16: kNN Accuracy Drop with Increasing k in Continuous Action Space Embeddings (left). Training loss for training action representations (right)

To assess the quality of the learned embeddings, we conduct an evaluation using a k-Nearest Neighbors (kNN) classifier. Specifically, we train kNN on a subset of the embeddings and analyze the impact of increasing k on test accuracy. The intuition behind this evaluation is that well-separated embeddings should be locally consistent, meaning that a small k (considering only close neighbors) should yield high accuracy, while increasing k (incorporating more distant neighbors) may introduce noise and reduce accuracy as shown in Fig 16.

### F.3 INTEGRATING VISUAL AND TEXTUAL REPRESENTATIONS

Incorporating a textual backbone alongside the image backbone yielded significantly lower loss values and faster convergence compared to using an image-only backbone.

This improvement can be attributed to several factors:

1. Semantic Guidance: Textual representations carry rich semantic information that can guide the image backbone. Instead of relying solely on visual cues, the model gains an additional perspective on the underlying concepts (e.g., object names, attributes, or relations).

2. Improved Discriminative Power: With access to text-based information, the model can differentiate between visually similar classes by leveraging linguistic differences in their corresponding textual descriptions.

3. Faster Convergence: Because textual features often come from large, pretrained language models, they are already highly informative. Injecting these features into the training pipeline accelerates the learning process, reducing the number of iterations needed to reach a satisfactory level of performance.

## G FINE-TUNING

Once failure modes are identified, we empirically found that the most effective strategy for fine-tuning the manipulation policy, $\pi^R$, is to use all selected failure samples together rather than iteratively adapting to subsets (Fig. 17). To adapt $\pi^R$ against identified failures, we target specific environment variations that a user wishes to improve. As shown in Table 13, fine-tuning with the high-likelihood samples provided by $\pi_{MD}$ yields the highest accuracy. We then *fine-tune* $\pi^R$ on the combined set of targeted samples, ensuring corrections for critical failures (Fig. 18). Notably, fine-tuning on a large set of failure modes also leads to accuracy improvements. When computational resources permit, fine-tuning on all identified failures may be also effective; however, when resources are constrained, leveraging RoboMD to select the most informative subset provides an efficient and robust strategy for policy adaptation.

### G.1 ADDITIONAL RESULTS

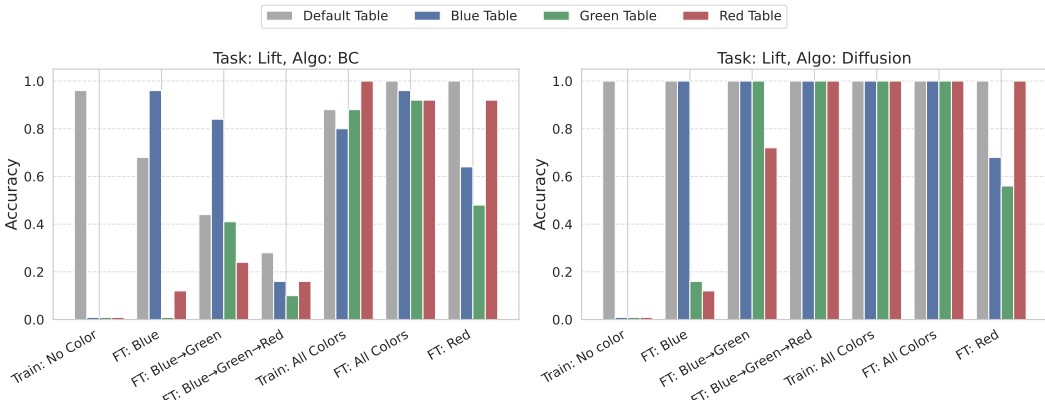

Figure 17: Performance comparison of behavior cloning (BC) and diffusion-based policies on the Lift task before and after fine-tuning with failure-inducing samples. Each bar represents the success rate of the policy across different **table colors**.

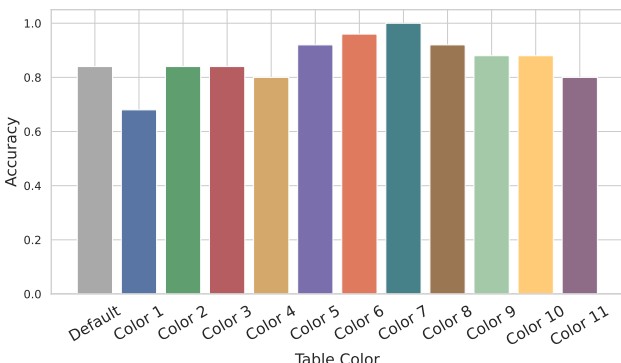

Figure 18: BC lift finetuned on a combined dataset of 12 different Table colors

Table 13: Evaluation of Fine-Tuning Approaches on Failure-Targeted Data.

| Fine-tuning (FT) Strategies | Accuracy %($\uparrow$) | Mean Square Distance ($\downarrow$) | Wasserstein Error ($\downarrow$) | Chi-Square Size | FT Dataset Size |
|---|---|---|---|---|---|
| Pre-trained | 67.91 | 0.377 | 0.005 | 0.016 | – |
| FT with RoboMD | **92.83** | **0.033** | **0.001** | **0.001** | 1.29GB |
| FT with 1 failure | 71.25 | 0.068 | 0.002 | 0.003 | 0.43GB |
| FT with 2 failure | 75.83 | 0.140 | 0.003 | 0.006 | 0.86GB |
| FT with 3 failure | 75.41 | 0.050 | 0.002 | 0.002 | 1.29GB |
| FT with 4 failure | 80.00 | 0.128 | 0.002 | 0.005 | 1.72GB |
| FT with 5 failure | 81.25 | 0.337 | 0.004 | 0.014 | 2.15GB |
| FT with 6 failure | 64.76 | 0.201 | 0.003 | 0.008 | 2.58GB |
| FT with all failure | 85.48 | 0.069 | 0.002 | 0.003 | 9.00GB |

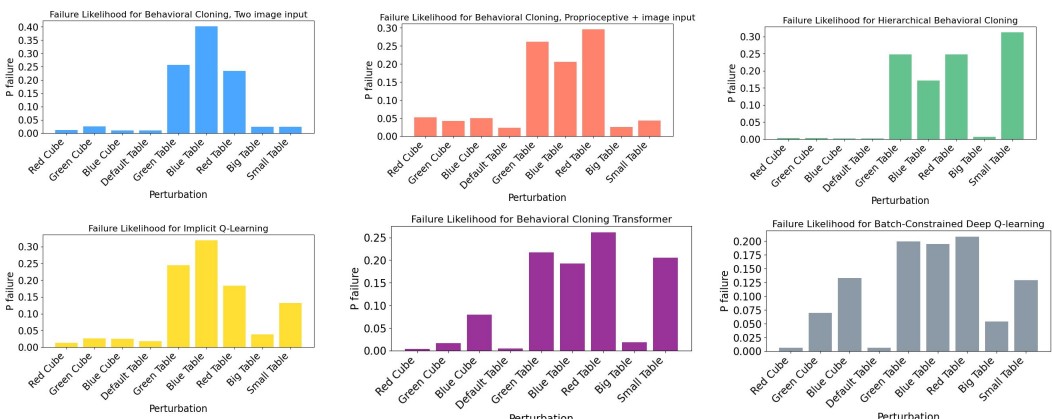

Figure 19: Comparison of model robustness by measuring failure likelihood under controlled small environmental variations.

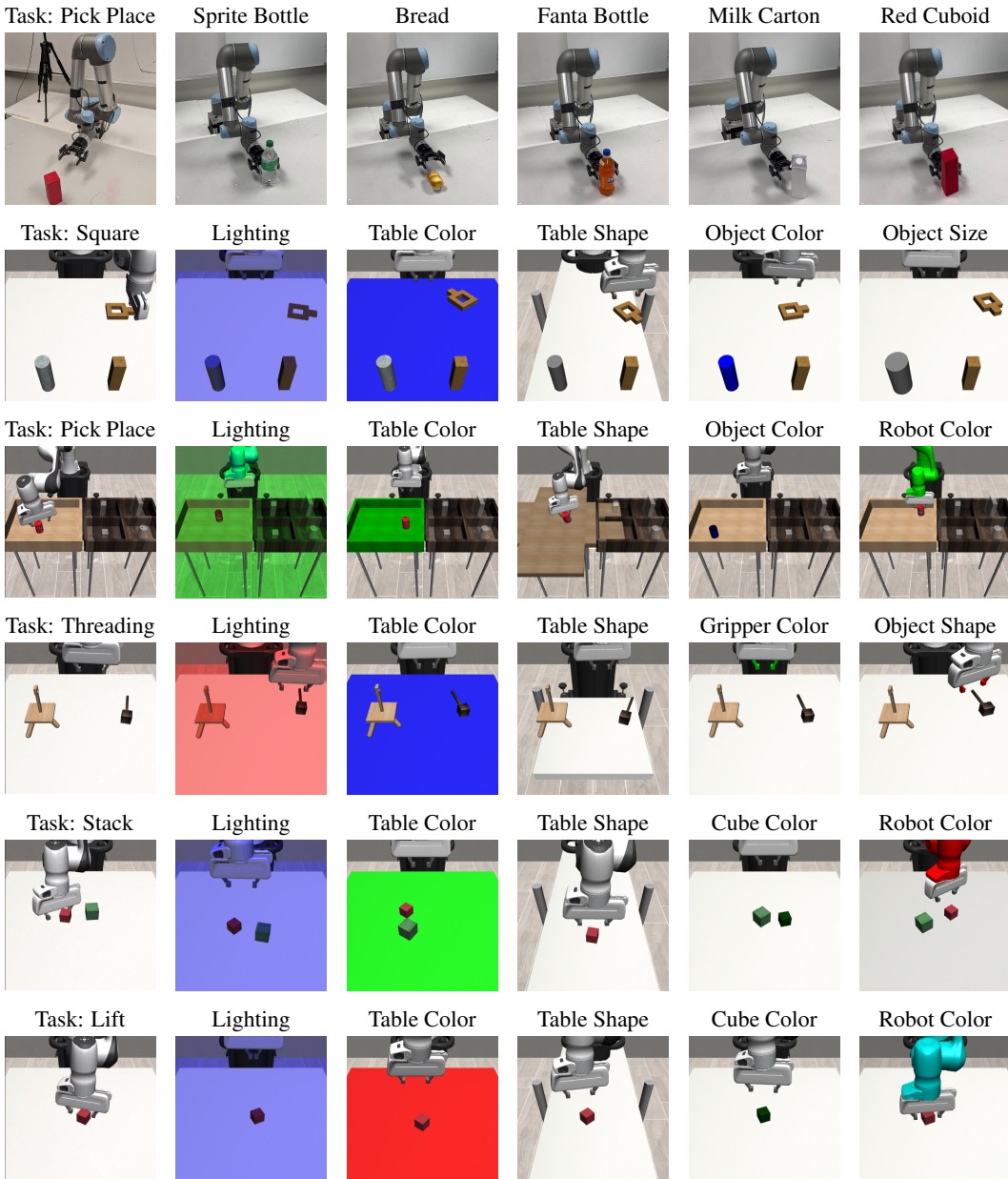

Figure 20: Environmental and Object Perturbations on Manipulation Tasks

