# OpenReview forum: "RoboMD: Uncovering Robot Vulnerabilities through Semantic Potential Fields"
_ICLR.cc/2026/Conference — ICLR 2026 Poster_

### Official Review · Reviewer_DxbL · 2025-10-24

**Soundness:** 3
**Presentation:** 3
**Contribution:** 3
**Rating:** 6
**Confidence:** 2

**Summary:**

This paper proposes a framework for predicting robot manipulation vulnerabilities by learning a deep RL policy over a vision–language embedding space. It identifies vulnerable regions without costly real-world testing, builds probabilistic vulnerability maps, and uncovers 23% more unique vulnerabilities than baselines, improving downstream robustness.

**Strengths:**

1. The paper tackles a highly practical and underexplored problem—diagnosing robot manipulation vulnerabilities—using a smart and scalable approach.

2. Reformulating exploration over a learned vision–language embedding is elegant and helps the policy find subtle failure modes efficiently.

3. The experimental evaluation is thorough, spanning both simulation and real-world settings, and clearly shows strong generalization and effectiveness.

**Weaknesses:**

1. The method doesn’t really explain why vulnerabilities emerge, so it feels more like a tool for finding problems rather than understanding them.

2. The embedding-based approach may not fully capture complex real-world dynamics, which could limit how well it scales to more unpredictable environments.

**Questions:**

How well would the proposed method handle scenarios where the embedding space fails to capture subtle but critical real-world variations?

Can the framework provide more interpretable insights into why specific vulnerabilities occur, rather than just locating them?

---

> ### Author Response · Authors · 2025-11-22
> **Response to reviewer's comments part 1/2**
>
> We sincerely thank the reviewer for their thoughtful assessment and for highlighting the strengths of our work, specifically the practicality of diagnosing robot vulnerabilities, and the elegance of reformulating exploration in a vision-language embedding space. Below we address all concerns raised :
> 1. **Concern 1 and Question 2** - Interpretability: Once failures are detected with RoboMD, existing post-hoc interpretability methods (e.g., TCAV, Integrated Gradients) can be used on top of it to explain the failures. We demonstrate that high-level reasoning is possible by inspecting RoboMD’s discovered failure modes.
> 2. **Concern 2 - Embedding Limitations**: RoboMD’s modular design allows easy integration of newer and more capable video–language models we also provide extensive quantitative analyses.
> 3. **Questions 1**: minor changes that do not alter task success naturally yield similar embeddings. The framework remains modular, stronger or task-specific representations can be swapped in seamlessly.
>
> Please refer to this [Anonymized GitHub Link](https://anonymous.4open.science/r/ICLR-Rebuttal-0A27/DxbL/Reviewer_Dxbl.md) where we have compiled detailed explanation and images/plots for better understanding.
>
> ## Concern 1 and Question 2: Interpretability
>
> We agree with the reviewer’s observation that RoboMD “finds problems rather than explaining why they occur,” and emphasize that this distinction is intentional, not a limitation.
>
> RoboMD is designed as a diagnostic tool whose purpose is to identify and localize failures efficiently in a continuous semantic space i.e., to answer where and when a manipulation policy fails or could fail. Causal or conceptual explanation answering why a failure occurs is a complementary step that can be layered on top of RoboMD’s outputs. This separation mirrors the standard paradigm in interpretable AI where detection precedes explanation. For example, once RoboMD pinpoints high-likelihood failure regions, concept-based interpretability methods such as TCAV [1], Integrated Gradients [2], Concept Bottleneck Models [3], BaTCAVe [4], can be applied to those same instances to reveal the human-interpretable causes of failure (e.g., lighting occlusion, object-color bias). [5] also demonstrates how multimodal reasoning can summarize failure causes once the vulnerable conditions are identified. RoboMD directly enables such analyses by providing a dense, semantically structured map of failure regions.
>
> Additionally, we would like to emphasize that RoboMD also enables high-level reasoning about the causes of failures by examining the failure modes it identifies. For instance, as shown in Figure 7 of the paper, RoboMD reveals that most failures occur due to changes in table color, as failure modes predominantly cluster around the {Red, Blue, Green} table settings. We further corroborate this with additional real-world experiments using the SO101 model, with its failure mode illustrated in [Image](https://anonymous.4open.science/r/ICLR-Rebuttal-0A27/DxbL/Images/real_robot_policy_side_by_side.png) from which we can infer that the most failure action and infer root case as shown below in Rebuttal Table 4.1.
>
> **Rebuttal Table 4.1:** Identified failure mode and explanation for different real world policies.
> | Model    | Failure Mode | High level explanation                              |
> |-----------|--------------|-----------------------------------------------------|
> | Smol vla  | Action 4     | Change in table color causes failure               |
> | ACT       | Action 5     | Change in table color causes failure               |
> | π0       | Action 5     | Change table color causes failure                  |
> | GR00T     | Action 5,7   | Change in table color or cube size causes failure  |
>
> ## Concern 2: Embedding Limitations
>
> We thank the reviewer for this valuable comment. RoboMD already incorporates a short sequence of frames, allowing it to implicitly learn temporal patterns up to a certain extent. As video–language encoders (e.g., Gemini Robotics, Video-LLaVA, VideoCLIP) continue to advance, these can be directly integrated into our modular framework to improve dynamic understanding. The strength of RoboMD lies in its plug-and-play design, stronger vision-language backbones can naturally enhance its failure detection capabilities without architectural changes.
>
> **Additional Result:**
> We further measured how physical variations map to embedding-space distances and show that embedding distances change monotonically with variation magnitude (e.g., lighting, object size, table color), achieving perfect or near-perfect correlations (τ = 1.0, r > 0.86). This confirms that RoboMD’s learned embeddings correspond to physically meaningful variations, ensuring reliable generalization within real-world limits.

---

> ### Author Response · Authors · 2025-11-22
> **Response to reviewer's comments part 2/2**
>
> **Rebuttal Figure 4.2:** Image links: [Left](https://anonymous.4open.science/r/ICLR-Rebuttal-0A27/DxbL/Images/intensity_transition_embedding_space.png), [Right](https://anonymous.4open.science/r/ICLR-Rebuttal-0A27/DxbL/Images/cumulative_distance_vs_red_intensity.png) Shows Link π0 embedding visualization (Left) t-SNE visualization of 5 rollouts projected into 2D embedding space. Points are colored by success rate (red = failure, green = success). (Right) Relationship between red light intensity and cumulative distance progression. Each demo (Demo 0–4) represents a step from high to low red intensity
>
> **Rebuttal Table 4.2.** Correlation analysis between physical variation and cumulative embedding distance.
>
> | Model    | Variations  | Kendall’s tau | Pearson’s r |
> |-----------|-------------|---------------|--------------|
> | BC Stack  | Cube size   | 1.000         | 0.924        |
> | BC Can    | Table color | 1.000         | 0.842        |
>
> These perfect or near-perfect correlations (τ = 1.0, r > 0.84) confirm that semantic embedding distances grow predictably with physical variation magnitude.
>
> ## Questions
>
> **Q1. Handling scenarios where the embedding fails to capture subtle real-world variations.**
> We thank the reviewer for this important question. In practice, RoboMD is generally able to capture subtle variations, as it is unlikely that a vision language model would produce identical embeddings for different real-world configurations due to its pixel-level sensitivity as shown in Rebuttal Table 4.2.
>
> **Rebuttal Table 4.3:** Cosine similarity between similar demos within our embedding space.
> All variation have the cube displaced by a delta = 0.1 cm.
>
> |         | Variation 1 | Variation 2 | Variation 3 | Variation 4 | Variation 5 |
> |----------|--------------|--------------|--------------|--------------|--------------|
> |Variation 1   | 1.0000 | 0.6118 | 0.8697 | 0.6464 | 0.8696 |
> | Variation 2   | 0.6118 | 1.0000 | 0.8265 | 0.6505 | 0.8264 |
> | Variation 3   | 0.8697 | 0.8265 | 1.0000 | 0.8270 | 0.8670 |
> | Variation 4   | 0.6464 | 0.6505 | 0.8270 | 1.0000 | 0.8270 |
> | Variation 5   | 0.8696 | 0.8264 | 0.8670 | 0.8270 | 1.0000 |
>
>
> However, we would like to highlight that If a subtle real world variation leads to the same embedding it is highly likely that the outputs for both environments would be similar. Which we can infer from the below table where we change intensity of yellow light applied to the environment and see that subtle changes in the environment do not change success outcomes (which is the most important aspect for RL i.e the reward it gets).
>
> **Rebuttal Table 4.3**: yellow light Intensity vs success rate for π0 and Smol VLA.
> | Intensity % | Success (π0) | Success (Smol VLA) |
> |--------------|----------------|--------------------|
> | 40           | 1              | 1                  |
> | 41           | 1              | 1                  |
> | 42           | 1              | 1                  |
> | 43           | 1              | 1                  |
> | 44           | 1              | 1                  |
> | 45           | 1              | 1                  |
>
>
>
>
> **References**
>
> 1. Kim, Been, Martin Wattenberg, Justin Gilmer, Carrie Cai, James Wexler, Fernanda Viégas, and Rory Sayres. “Interpretability Beyond Feature Attribution: Quantitative Testing with Concept Activation Vectors (TCAV).” In Proceedings of the 35th International Conference on Machine Learning (ICML), pp. 2673–2682. 2018.
> 2. Sundararajan, Mukund, Ankur Taly, and Qiqi Yan. “Axiomatic Attribution for Deep Networks.” In Proceedings of the 34th International Conference on Machine Learning (ICML), pp. 3319–3328. 2017.
> 3. Koh, Pang Wei, Thao Nguyen, Yew Siang Tang, Stephen Mussmann, Emma Pierson, Been Kim, and Percy Liang. “Concept Bottleneck Models.” In Proceedings of the 37th International Conference on Machine Learning (ICML), pp. 5338–5348. 2020.
> 4. Sagar, Som, et al. "Trustworthy Conceptual Explanations for Neural Networks in Robot Decision-Making." arXiv preprint arXiv:2409.10733 (2024).
> 5. Liu, Zeyi, Arpit Bahety, and Shuran Song. “REFLECT: Summarizing Robot Experiences for Failure Explanation and Correction.” In IEEE International Conference on Robotics and Automation (ICRA), 2023.

---

> > ### Comment · Reviewer_DxbL · 2025-11-26
> >
> > Thank you for the author's reply. I have carefully reviewed the author's reply as well as the opinions of other reviewers. I have decided to maintain my score.

---

> > > ### Author Response · Authors · 2025-11-26
> > >
> > > Thank you for going through the rebuttal and providing us a response. We believe that we have effectively addressed all the reviewer's concerns through our detailed response, new real-world experiments with SOTA VLAs, and additional analyses.
> > >
> > > If there are any further specific concernes that is holding the reviewer back from increasing the rating, we would be very happy to provide it immediately before the deadline. Once again, thank you for supporting our research.

---

### Official Review · Reviewer_5NX2 · 2025-10-31

**Soundness:** 3
**Presentation:** 4
**Contribution:** 3
**Rating:** 4
**Confidence:** 4

**Summary:**

This paper presents RoboMD, a novel framework that diagnoses failure modes in robot manipulation policies by training a RL agent to explore a continuous vision–language embedding space. Rather than relying on hand-crafted test conditions or costly physical rollouts, RoboMD learns to search for vulnerabilities virtually by interpreting the embedding space as a semantic potential field that repels from known successes and attracts toward failures.

The core contributions include:

1. A deep RL-based diagnostic agent operating in a semantically structured latent space.

2. A framework that identifies both seen and unseen failure modes, improving upon vision-language models (VLMs) and heuristic baselines.

3. The use of identified vulnerabilities for targeted fine-tuning, leading to substantial improvements in robot manipulation policies.

Experiments are conducted on both simulated environments  and real-world scenarios (UR5e arm), demonstrating improved diagnostic performance and policy robustness.

**Strengths:**

### Originality
- Reformulation of diagnosis as RL search in a semantically meaningful potential field is elegant and novel.
- Using multimodal embeddings to structure the search space is a strong innovation.
### Quality
- Broad evaluation: 4 manipulation tasks × multiple policy types × real vs. simulated settings.
- Benchmarks vs. GPT-4o, Gemini, Qwen2-VL show strong generalization and robustness.
### Clarity
- Excellent visuals and modular structuring.
- Detailed Appendices with ablation studies and embedding quality analysis.
### Significance
- Addresses core robotic AI challenges: cost and risk of real-world rollouts, poor generalization, and hidden failure modes.
- Offers safe, scalable, and data-efficient diagnosis that’s practical for deployment-stage validation.

**Weaknesses:**

This work is of great practical significance, but I am allowed to express some concerns.
### Assumption of Embedding Quality:
1. The success of the approach hinges critically on the quality of the semantic embedding space. If poorly structured, RL search may yield meaningless trajectories.
2. Though addressed via BCE + contrastive loss, there is no quantitative measure of embedding smoothness across multiple datasets.
### Limited Real-World Testing:
1. While real-world results (UR5e) are included, most findings are in simulation.
2. Have not seen some of sim2real's settings. The community has always believed that real machine data is the most reliable, although simulation data has the advantage of large data and low cost. This work would have more relevance if real machine data was included in the loop.

**Questions:**

1. While the authors acknowledge the combinatorial complexity of real-world disturbances, the training and search process still relies on a limited and predefined set of action embeddings (either sampled or pre-collected). This raises a potential contradiction: how well does RoboMD generalize to more open-ended or compositional disturbances beyond the known embedding space? It would strengthen the paper to explicitly discuss the coverage limitations of the action space and possible mechanisms to expand it during deployment.

2. Several recent works [1][2] have begun to explore failure prediction in robot foundation models, especially through uncertainty-aware approaches. However, this paper does not compare against those methods or discuss how RoboMD relates to them. Some of these prior efforts incorporate uncertainty modeling (e.g., epistemic or aleatoric uncertainty) into their optimization process, which allows the model to focus exploration or calibration in regions of high uncertainty. I believe RoboMD would benefit significantly from integrating a similar mechanism. For example, encouraging the RL agent to focus its exploration around uncertain or ambiguous regions in the semantic space. If the authors can demonstrate that such uncertainty-aware exploration improves the agent’s performance or failure detection coverage, it would substantially increase my confidence in the generality and practical robustness of this approach.


3. The framework would benefit from a demonstration of real-world deployment, even at small scale. A promising direction is to implement a sim-to-real flywheel loop, where $\pi_{MD}$  is first trained in latent space using synthetic or simulation-aligned embeddings, then deployed to identify high-risk conditions on a physical robot. The collected failure cases could be added to the embedding space and used to retrain $\pi_{MD}$  in future iterations. This would close the loop and demonstrate practical deployability, which currently remains underexplored.

> [1] SAFE: Multitask Failure Detection for Vision-Language-Action Models Qiao Gu etc. Neurips 2025
> [2] Can We Detect Failures Without Failure Data? Uncertainty-Aware Runtime Failure Detection for Imitation Learning Policies Chen Xu etc. CoRL 2024

---

> ### Author Response · Authors · 2025-11-22
> **Response to reviewer's comments part 1/4**
>
> We sincerely thank the reviewer for their detailed and thoughtful evaluation. We are grateful for the reviewer’s observation that RoboMD addresses a key gap in robot safety: enabling scalable, data-efficient, and deployment-ready failure analysis without costly or unsafe physical rollouts. These points align closely with our goals, and we appreciate the reviewer’s constructive feedback, which helped us further strengthen the paper through additional quantitative embedding analyses, new real-world validation, and expanded discussion on uncertainty-aware exploration. Below we summarize our responses:
> 1. **Concern 1 - Quantitative Embedding Smoothness and Structure**: We show new quantitative analyses showing correlation between physical semantic variations and embedding distances (τ = 1.0, r > 0.9).
> 2. **Concern 2 - Limited Real-World Testing**: We expanded real-world validation with four distinct SOTA VLA policies (ACT, SmolVLA, π0, GR00T),
> 3. **Concern 3 - sim-to-real**: RoboMD reduces reliance on traditional sim-to-real transfer by operating in a semantic embedding space. It can be trained and deployed directly on real robots with minimal overhead, serving as a continuous diagnostic tool that adapts to new vulnerabilities from real-world data.
> 4. **Questions**: RoboMD generalizes through its semantic embedding–based RL search, providing measurable generalization boundaries via embedding distances. It is complementary to uncertainty-aware methods and can integrate such signals in future work. Real-world validations with multiple SOTA policies already demonstrate the framework’s practicality, forming the foundation for future sim-to-real extensions.
>
> Please refer to this [Anonymized GitHub Link](https://anonymous.4open.science/r/ICLR-Rebuttal-0A27/SeQx/Reviewer_SeQx.md) where we have compiled detailed explanation and images/plots for better understanding.
>
> ## Concern 1.1: Embedding smoothness
>
> We appreciate the reviewer's emphasis on providing quantitative evidence for embedding smoothness. Our approach employs an MSE objective with L2 regularization (Eq. 1), which theoretically promotes smoothness in the learned representation [1]. Below we present empirical validation of this property through systematic analysis.
>
> **Experimental Methodology:** To assess smoothness, we systematically modify physical scene properties (illumination conditions, object appearance, geometric scale) and track corresponding distance metrics within the embedding manifold.
>
> **Quantitative Analysis:** Rebuttal Tables 3.1 and 3.2 demonstrate a strong monotonic relationship between physical parameter changes and embedding-space distances. Specifically, we observe perfect rank correlation (Kendall's τ = 1.0) and high linear correlation (Pearson's r > 0.86), establishing that embedding distances faithfully capture semantic variations in the physical domain.
>
> Figure 3.1 illustrates this relationship for Pi0 and SmolVLA policies, depicting gradual transitions from failure regions (red) to success regions (green) as we reduce lighting intensity.
>
> **Rebuttal Table 3.1.** Cumulative distance from the initial demonstration increases monotonically with decreasing lighting intensity for the Pi0 policy.
>
> | Red Light Intensity              | Policy Rollouts | Cumulative Distance in Embedding |
> |----------------------------------|------------------|----------------------------------|
> | Red = 255 (completely red)       | Failed           | 0.00                             |
> | Red = 235                        | Failed           | 0.22                             |
> | Red = 187                        | Succeed          | 0.55                             |
> | Red = 163                        | Succeed          | 0.58                             |
> | Red = 140 (light pink)           | Succeed          | 0.67                             |
>
> Link to Figure [Image1](https://anonymous.4open.science/r/ICLR-Rebuttal-0A27/DxbL/Images/intensity_transition_embedding_space.png), [Image2](https://anonymous.4open.science/r/ICLR-Rebuttal-0A27/DxbL/Images/cumulative_distance_vs_red_intensity.png)
> **Fig 3.1:** Shows Pi0 embedding visualization (Left) t-SNE visualization of 5 rollouts projected into 2D embedding space. Points are colored by success rate (red = failure, green = success). (Right) Relationship between red light intensity and cumulative distance progression.
>
> **Rebuttal Table 3.2.**
> Correlation analysis between physical variation and cumulative embedding distance.
>
> | Model    | Variations  | Kendall's τ | Pearson's r |
> |-----------|--------------|--------------|--------------|
> | BC Stack  | Cube size    | 1.000        | 0.924        |
> | BC Can    | Table color  | 1.000        | 0.842        |
>
> The observed correlations (τ = 1.0, r > 0.84) validate that embedding distances scale proportionally with the magnitude of physical variations, indicating a well-structured and predictable embedding manifold.

---

> ### Author Response · Authors · 2025-11-22
> **Response to reviewer's comments part 2/4**
>
> **Numerical Assessment:** To evaluate local smoothness, we compute pairwise distances between adjacent embeddings (e_i and e_{i+1}) within failure and success regions. This metric quantifies the rate of change across the embedding space, ensuring that the learned manifold exhibits sufficient regularity for effective policy optimization. Low L2 distances (||e1 – e2||_2) between consecutive embeddings signal that the representation maintains continuity.
>
> **Rebuttal Table 3.3.**
> Continuity analysis (d₁[i] = embedding[i+1] – embedding[i]) across policies.
> | Policy       | Mean \|d₁ − d₂\| |
> |---------------|------------------|
> | Smol VLA [2]  | 0.267 ± 0.367    |
> | GR00T [3]     | 0.207 ± 0.397    |
> | π0 [4]      | 0.282 ± 0.359    |
> | ACT [5]       | 0.273 ± 0.364    |
>
> The consistent magnitude of these distances across different policies provides evidence that RoboMD learns a continuous, well-behaved embedding space that preserves semantic structure while maintaining numerical smoothness.
>
> ## Concern 1.2: Embedding Structure
>
> We evaluate embedding structure through a comprehensive four-part analysis: (1) assessing clustering quality within success/failure regions, (2) examining transition smoothness across region boundaries, (3) verifying correspondence with semantic-visual representations, and (4) establishing monotonic relationships with physical variations.
>
> We systematically evaluated embedding quality across failure and success regions (corresponding to actions/dotted lines in Fig 4 from the paper) for multiple manipulation policies and tasks, including expanded real-world experiments with SOTA vision-language-action models.
>
> **Rebuttal Table 3.4.** Embedding-space grouping metrics showing strong separation and smoothness across tasks and policies, validating the embedding's semantic consistency.
>
> | Domain       | Policy        | Separation Ratio (>1.5 indicates good) | Effect Size (Cohen's d) (>0.8 indicates good) | Silhouette Score (>0.5 indicates good grouping) | Davis–Bouldin Index (<1.0 indicates good) |
> |---------------|----------------|-----------------------------------------|-----------------------------------------------|--------------------------------------------------|---------------------------------------------|
> | Simulation    | BC – Lift       | 22.993                                  | 2.920                                         | 0.907                                            | 0.149                                       |
> | Simulation    | BC – Square     | 20.232                                  | 1.922                                         | 0.875                                            | 0.245                                       |
> | Real Robot    | SmolVLA [2]     | 10.371                                  | 4.522                                         | 0.871                                            | 0.347                                       |
> | Real Robot    | ACT [3]         | 7.760                                   | 4.767                                         | 0.839                                            | 0.384                                       |
> | Real Robot    | π0 [4]         | 23.974                                  | 4.212                                         | 0.953                                            | 0.068                                       |
> | Real Robot    | GR00T [5]       | 4.261                                   | 3.309                                         | 0.701                                            | 0.533                                       |
>
> **Metric Interpretations (Rebuttal Table 3.4):**
> 1. **Separation Ratio:** Defined as the ratio of average inter-region distance to average intra-region distance. Our results show values ranging from 4.26 to 23.97, all exceeding the 1.5 threshold, which confirms that failure modes cluster into clearly separated regions within the embedding space.
> 2. **Effect Size (Cohen's d):** Quantifies the standardized magnitude of difference between regions using pooled standard deviation. All tasks exhibit large effect sizes (1.92-4.77), providing strong evidence for meaningful separation between failure mode clusters.
> 3. **Silhouette Score:** Evaluates how well embeddings match their assigned region relative to neighboring regions. Scores above 0.70 (range: 0.70-0.95) across all tasks indicate high within-region cohesion.
> 4. **Davis-Bouldin Index:** Computes the ratio of within-cluster dispersion to between-cluster separation, where lower values indicate better clustering. All tasks achieve scores below 0.53 (range: 0.068-0.533), demonstrating excellent separation quality.
>
> For transitions between regions, we investigate two critical aspects in the next comment: (1) robustness of interpolated embeddings to local perturbations, ensuring they remain within their respective regions, and (2) correspondence between embedding-space distances and semantic-visual feature distances.

---

> ### Author Response · Authors · 2025-11-22
> **Response to reviewer's comments part 3/4**
>
> **Rebuttal Table 3.5:**
> Quantitative evaluation of embedding-space continuity. Local smoothness shows if interpolated points are still close to the initial region. Cross-Modal Action Alignment captures correlation between embedding-space distances and semantic-visual distances (CLIP text + image features).
> | Policy   | Local Smoothness (penalty up to 15) | Cross-Modal Action Alignment |
> |-----------|-------------------------------------|-------------------------------|
> | ACT       | 93.4%                              | 58.9%                         |
> | Smol VLA  | 92.9%                              | 57.0%                         |
> | π0      | 94.3%                              | 57.9%                         |
>
> **Metric Interpretations (Rebuttal Table 3.5):**
> 1. **Local Smoothness:** Evaluated by introducing noise to embeddings and verifying that perturbed embeddings remain within the same failure mode region. Scores exceeding 80% indicate highly smooth local structure, which our results consistently achieve (92.9-94.3%).
> 2. **Cross-Modal Action Alignment:** Calculated as cosine similarity between embedding-space distance vectors and semantic-visual distance vectors (derived from CLIP features of image-text pairs) across action pairs. Our observed values (55-60%) represent an optimal balance:
>     - Complete alignment (100%) would suggest the embedding provides no task-specific information beyond generic semantic features.
>     - No alignment (0%) would indicate the embedding fails to capture semantic relationships.
>
> **References**
> 1. Bottou, Léon,, et all "Optimization methods for large-scale machine learning." SIAM review 60.2 (2018): 223-311.
> 2. Shukor, Mustafa, et al. "Smolvla: A vision-language-action model for affordable and efficient robotics." arXiv preprint arXiv:2506.01844 (2025).
> 3. Zhao, Tony Z., et al. "Learning fine-grained bimanual manipulation with low-cost hardware." arXiv preprint arXiv:2304.13705 (2023).
> 4. Black, Kevin, et al. "π0: A vision-language-action flow model for general robot control. CoRR, abs/2410.24164, 2024. doi: 10.48550." arXiv preprint ARXIV.2410.24164.
> 5. Bjorck, Johan, et al. "Gr00t n1: An open foundation model for generalist humanoid robots." arXiv preprint arXiv:2503.14734 (2025).
>
> ## Concern 2: Limited Real-World Testing
> While RoboMD is platform agnostic, we have now conducted extensive real-world experiments on SOTA VLAs with a new real world SO101 robot manipulator, significantly expanding our validation:
> 1. **4 New SOTA real world Policies**: We finetuned and evaluated ACT, Smol VLA, Pi0, and GR00T policies on the SO101 robot, collecting 35 demos per policy across 7 distinct failure modes.
> 2. We tested unseen environment ranking on action (purple lighting) not present in Table 3.7. RoboMD currently predicts in π0 and SmolVLA it works > red light (failure) and in GR00T < red light (failure) which matches the ground truth where purple light worked in π0 and SmolVLA and not in GR00T. This result will be added to Table 3 of the paper.
> 3. Comprehensive Failure Detection Analysis: We collected detailed failure detection data across all policies, revealing distinct, policy-specific failure patterns:
>
> The plots for failure modes of the four real world policies are provided here : [Image] (https://anonymous.4open.science/r/ICLR-Rebuttal-0A27/SeQx/Images/real_robot_policy_side_by_side.png)
>
> ## Concern 3 and Question 3: Sim-to-Real Loop
> We thank the reviewer for highlighting the importance of sim-to-real validation and real-world deployment. We fully agree that real machine data provides the most reliable measure of robustness. While we have not yet implemented a fully automated sim-to-real loop, RoboMD has already been validated on multiple real-robot platforms (Pi0, GR00T, SmolVLA, and ACT on the SO101 arm), demonstrating the practicality of our diagnostic framework.
> Historically, sim-to-real was critical due to the difficulty of collecting real robot data, but recent trends show that many modern robotic systems now train and fine-tune directly on real hardware seeded from large pretrained models. However, as the reviewers suggest, we hope to do more *engineering efforts* to implement the sim2real and open source as future work.
>
> ## Questions:
> **Q1: How well does RoboMD generalize to more open-ended or compositional disturbances?**
> RoboMD generalizes to unseen or compositional disturbances through its semantic embedding–driven RL search, which enables exploration across continuous variations sharing semantic similarity with known failures. While we acknowledge that entirely novel, out-of-semantic disturbances cannot be predicted—a limitation common to all data-driven systems, RoboMD provides a measurable boundary of generalization using embedding-space distance metrics. Furthermore, we analyzed the penalty term in our reward formulation and observed that for highly complex, unseen environments with multiple distractors, the penalty value rises sharply (often >10).

---

> ### Author Response · Authors · 2025-11-22
> **Response to reviewer's comments part 4/4**
>
> **Q2: How does RoboMD relate to uncertainty-aware methods?**
>
> We see three ways to detect failures:
> 1. Method 1: Building a likelihood of success with only successful trials [6] - when an unfamiliar input comes, this method will flag it. As [6] nicely shows, this is useful in runtime monitoring so that the human can intervene.
> 2. Method 2: Building a classifier with success and failure data, similar to [7] which will be presented next month at Neurips. We have included results for a naive version of this approach in Rebuttal Table 3.6.2. As the reviewer has requested, we have now included results of methods 1 and 2 by incorporating uncertainty as well as shown in Rebuttal Table 3.6.1 and 3.6.2.
> 3. Method 3: Searching for failures - In our method, we get an embedding from method 2 and search for failures further using deep RL. Incorporating uncertainty into method 2 and employing Bayesian RL could, in principle, further improve performance beyond our approach, and we encourage future work to explore this direction.
>
>
> **Rebuttal Table 3.6.1.** Models trained with only success data and thresholding (Concern 2 method 1).
>
> | Task   | Our Best | MC Dropout (0.5) CNN | MC Dropout (0.5) ResNet | Ensemble CNN | Ensemble ResNet | Ensemble CNN+ResNet |
> |--------|-----------|------------------------|---------------------------|----------------|------------------|----------------------|
> | Lift   | 82.30     | 46.95                 | 37.17                    | 45.31          | 36.10            | 37.35               |
> | Can    | 84.00     | 48.27                 | 46.99                    | 45.62          | 43.13            | 41.18               |
> | Square | 76.00     | 50.22                 | 46.30                    | 49.32          | 49.62            | 50.57               |
>
> **Rebuttal Table 3.6.2.** Models trained with both success and training data (Concern 2 method 2).
>
> | Task   | CNN | ResNet | MC Dropout (0.5) CNN | MC Dropout (0.5) ResNet | Ensemble CNN | Ensemble ResNet |
> |--------|------|---------|------------------------|---------------------------|----------------|------------------|
> | Lift   | 32.72 | 51.17  | 38.91                 | 51.23                    | 36.76          | 55.00            |
> | Can    | 35.44 | 52.25  | 37.94                 | 53.32                    | 35.27          | 48.82            |
> | Square | 14.00 | 47.00  | 21.16                 | 48.22                    | 12.92          | 43.95            |
>
> **Rebuttal Table 3.6.3.**  Conformal Prediction and Stochastic Variational Inference.
>
> | Task   | Conformal Prediction CNN | Conformal Prediction ResNet | Stochastic Variational Inference CNN | Stochastic Variational Inference ResNet |
> |--------|----------------------------|------------------------------|---------------------------------------|------------------------------------------|
> | Lift   | 25.15                     | 58.09                        | 50.88                                 | 41.97                                    |
> | Can    | 30.65                     | 55.37                        | 47.82                                 | 37.67                                    |
> | Square | 9.97                      | 53.28                        | 48.28                                 | 49.14                                    |
>
> We would also like to clarify that conceptually, RoboMD differs from the said baselines as follows:
> 1. **Paradigm difference**: Uncertainty-based methods perform runtime failure detection during policy execution, while RoboMD performs offline vulnerability discovery before deployment. These are complementary but serve different purposes.
> We would also like to highlight that **RoboMD effectively identifies failure modes that serve as highly informative samples for fine-tuning**. By selecting these targeted failure cases, the model achieves better performance after fine-tuning compared to using larger but less focused datasets as shown in Table 5 of the paper.
> 2. Integration potential: Uncertainty-based methods could be integrated with RoboMD for enhanced performance, uncertainty estimates could guide exploration, and RoboMD-discovered vulnerabilities could improve uncertainty calibration. In such cases, RL will operate on a space of distributions rather than on a vector. While there seems to be some recent interest in exploring such directions[8], currently, to the best of our knowledge, it is impossible to perform PPO on distributional states.
>
> **References**
> 6. Can We Detect Failures Without Failure Data? Uncertainty-Aware Runtime Failure Detection for Imitation Learning Policies Chen Xu etc. CoRL 2024
> 7. SAFE: Multitask Failure Detection for Vision-Language-Action Models Qiao Gu etc. Neurips 2025
> 8. Tao, Ruo Yu, et al. "Benchmarking Partial Observability in Reinforcement Learning with a Suite of Memory-Improvable Domains." arXiv preprint arXiv:2508.00046 (2025).

---

> > ### Author Response · Authors · 2025-11-26
> >
> > We truly hope we answered your primary concerns regarding "Limited Real-World Testing" and "Embedding Quality" by conducting new real world experiments with 4 SOTA policies and providing the requested quantitative smoothness analysis.
> >
> > Given these substantial additions and new results, we believe we have fully responded to all of your questions. We would greatly appreciate it if you would consider raising your score to support our work. As the rebuttal period is ending soon, please let us know if there are any further clarifications we can provide.
> >
> > Once again, thank you for your time and for supporting our research.

---

### Official Review · Reviewer_SeQx · 2025-10-31

**Soundness:** 2
**Presentation:** 3
**Contribution:** 2
**Rating:** 6
**Confidence:** 4

**Summary:**

The paper considers robotic manipulation and preventing mistakes or ‘vulnerabilities’ when using large model vision-language (VLM)  perception, that may include variations in object properties.  Starting with a trained manipulation policy, a new RL process is trained over variations to find such vulnerabilities, called RoboMD. This policy can be queried to predict a failure over a sequence of actions, and given a finite set of failures can put probabilities on these forms. These are then used to fine tune the original manipulation policy.  Experiments are carried out in RoboSuite and show physical plus simulation experiments, and address some key issues.

**Strengths:**

The idea of creating a failure predictor is a good one, as can be used in many dynamic settings.  This can identify ‘distance’ from key vulnerabilities, and this can include lighting variation and object color and size (cases that are notorious for their difficulty to handle generally).

The method can be used to isolate and avoid some key failure modes. With sufficient prior examples, generalization over virtual roll outs is carried out.

The theory shows how a shaped MDP will lead to efficient training.

While depending on prior knowledge and setting, the method might be applicable for a variety of settings. The method shows some ability to generalize, given a sufficient failure training set.

The authors provide code and overall the work has good reproducibility.  This work has a good probability of leading to other follow on efforts.

**Weaknesses:**

The primary weakness is lack of adequate baselines.  There are various approaches for control that provide safety, such as barrier methods. It is not surprising that a conventional RL would not do as well as one supplemented with an additional failure mode predictor. The semantic potential field amounts to a kind of semantic barrier.  Also not clear is what happens to a vanilla RL if it is given some explicit bad examples with negative reward.

The method keys on the training selection and variations, so this is a critical step that relies on the user, and while generalization is shown in the examples, it isn't clear how this translates to new or more complex environments.

The term ‘environmental variation’ might be too general, since it appears it is more about sizing objects, color of objects, and lighting variations.

The number of predetermined versus virtual rollouts is a key question. It was not easy to see how this was precisely addressed in the experiments.

The method can identify some key failure modes, although guarantees are not clear. In the end we are still restricted by the VLM-based approach that doesn’t inherently do well with spatial reasoning.

The 3 theorems are interesting, showing how the MDP approach can lead to a shaped MDP that leads to efficient exploration and learning convergence.  However, these don’t quite address the key questions of covering the vulnerability space, or semantic related issues in general.

**Questions:**

How can the predetermined bad cases be incorporated into a vanilla RL approach?

What is the tradeoff in predetermined examples versus virtual samples?

What further experiments can be made to compare with other control safety methods?

---

> ### Author Response · Authors · 2025-11-22
> **Response to reviewer's comments part 1/3**
>
> We sincerely thank the reviewer for their thoughtful feedback. We greatly appreciate the recognition of our central idea of formulating failure prediction as a reinforcement-learning process in a semantic potential field and the observation that this approach can generalize across diverse visual and physical variations. We are also glad that the reviewer recognized its potential for broader impact and follow-on work. Below we address all concerns raised:
> 1. **Concern 1** - Missing Baselines and vanilla RL: Discussed relationship to CBF. Added uncertainty-based, conformal prediction and energy model baselines, also explained how vanilla RL  is a special case when all variations are known.
> 2. **Concern 2** - Generalization to New Environments: Demonstrated robustness on four new real-world policies (ACT, SmolVLA, π0, GR00T).
> 3. **Concern 3** - Definition of Environmental Variation: Defined variations explicitly. Real-world tests included added distractors for greater complexity.
> 4. **Concern 4** - Predetermined vs Virtual Rollouts: Clarified how rollouts worked and. outlined the tradeoffs
> 5. **Concern 5** - Recent VLMs such as Gemini Robotics-ER 1.5 [6] show strong spatial and geometric reasoning, RoboMD’s modular backbone can seamlessly integrate such advances, further enhancing spatial reasoning and failure detection.
> 6. **Concern 6** - Our three theorems focus on learning efficiency rather than classical control safety, showing that the potential-field MDP (1) preserves optimality, (2) accelerates convergence, and (3) promotes efficient exploration near success–failure boundaries.
>
> Please refer to this [Anonymized GitHub Link](https://anonymous.4open.science/r/ICLR-Rebuttal-0A27/SeQx/Reviewer_SeQx.md) where we have compiled detailed explanation and images/plots for better understanding.
>
> ## Concern 1 and Question 1: Missing Baselines (Barrier Methods, Vanilla RL with Bad Examples)
>
> We appreciate the reviewer for highlighting the connection to barrier methods, rooted in optimization and widely used in control theory, since such links are rarely discussed in ML conferences. To summarize the distinction succinctly:
> 1. ML approach (e.g., ours): Given images + text, does this look like it is going to fail?
> 2. Control Barrier Function approach: Given the safe set **and system dynamics**, is it mathematically impossible to remain safe?
>
> Barrier methods unquestionably offer formal safety certificates when the system dynamics are known. However, our setting is vision-to-action manipulation, where the true dynamics are not known. For this reason, classical CBF-based safety analysis is not directly applicable. Motivated by this distinction, we have expanded our set of SOTA baselines (i.e., newest VLAs published in 2025) accordingly to better reflect methods that operate without access to dynamics. We will surely highlight the link to barrier methods in the final version. While we found this paper [1], relevant to our discussion, please feel free to point us to any additional references that further bridge control-theoretic barrier methods with learning-based vulnerability analysis.
>
> To provide a more comprehensive comparison, we have incorporated additional baselines that are well-established in control theory and uncertainty quantification: Monte Carlo Dropout (which provides a Bayesian approximation to Gaussian processes [2] and is widely used in control applications), conformal prediction (a distribution-free uncertainty quantification method gaining traction in control theory), and variational inference (which has theoretical connections to path integrals and stochastic optimal control). Our benchmark results for these methods are presented below:
>
> **Rebuttal Table 2.1.1.**
> Accuracy report of models trained with only success data and thresholding (Concern 2 method 1).
>
> | Task   | Our Best | MC Dropout (0.5) CNN | MC Dropout (0.5) ResNet | Ensemble CNN | Ensemble ResNet | Ensemble CNN+ResNet |
> |--------|-----------|------------------------|---------------------------|----------------|------------------|----------------------|
> | Lift   | 82.30     | 46.95                 | 37.17                    | 45.31          | 36.10            | 37.35               |
> | Can    | 84.00     | 48.27                 | 46.99                    | 45.62          | 43.13            | 41.18               |
> | Square | 76.00     | 50.22                 | 46.30                    | 49.32          | 49.62            | 50.57               |
>
> Rebuttal Table 2.1.2 and 2.1.3 are in the next part

---

> ### Author Response · Authors · 2025-11-22
> **Response to reviewer's comments part 2/3**
>
> **Rebuttal Table 2.1.2.**
> Accuracy report of models trained with both success and training data (Concern 2 method 2).
>
> | Task   | CNN | ResNet | MC Dropout (0.5) CNN | MC Dropout (0.5) ResNet | Ensemble CNN | Ensemble ResNet |
> |--------|------|---------|------------------------|---------------------------|----------------|------------------|
> | Lift   | 32.72 | 51.17  | 38.91                 | 51.23                    | 36.76          | 55.00            |
> | Can    | 35.44 | 52.25  | 37.94                 | 53.32                    | 35.27          | 48.82            |
> | Square | 14.00 | 47.00  | 21.16                 | 48.22                    | 12.92          | 43.95            |
>
>
> **Rebuttal Table 2.1.3.**
> Accuracy report of models trained with Conformal Prediction and Stochastic Variational Inference.
>
> | Task   | Conformal Prediction CNN | Conformal Prediction ResNet | Stochastic Variational Inference CNN | Stochastic Variational Inference ResNet |
> |--------|----------------------------|------------------------------|---------------------------------------|------------------------------------------|
> | Lift   | 25.15                     | 58.09                        | 50.88                                 | 41.97                                    |
> | Can    | 30.65                     | 55.37                        | 47.82                                 | 37.67                                    |
> | Square | 9.97                      | 53.28                        | 48.28                                 | 49.14                                    |
>
>
> 2. **Vanilla RL with bad examples (special case)**: In our paper, we address this in Section 3.3 as a special case when all candidate variations are known priori:
>     - Action space: When variations are explicitly given (e.g., from historical failures or expert knowledge), RoboMD searches over A (a discrete set of known variations)
>     - The reward function assigns a negative scalar for success and a positive scalar for failure (as stated in Section 3.3)
>     - Exploration: RoboMD applies a finite sequence of predefined actions (a1, a2, ..., an) until a failure is induced
>     - This special case only works when all variations are known and can be explicitly enumerated. When variations are unknown, this approach cannot explore beyond the discrete set of known variations.
>
> **References**
> 1. Ballotta, Luca, et al. "Fault detection via output-based barrier functions." European Journal of Control (2025): 101283.
> 2. Gal, Yarin, and Zoubin Ghahramani. "Dropout as a bayesian approximation: Representing model uncertainty in deep learning." international conference on machine learning. PMLR, 2016.
>
> ## Concern 2: Generalization to New Environments
>
> Genaralization to complex env We thank the reviewer for this observation. While no method can fully predict outcomes in entirely new arbitrary complex environments, RoboMD is designed to generalize effectively within a domain given sufficient representative variations. This capability arises from the pre-trained vision-language semantic embedding, which leverages large-scale vision-language models (VLMs) trained on large diverse multimodal data. RoboMD’s embedding-space formulation enables continuous representation of unseen configurations, allowing robust generalization to new but semantically related conditions. For example, when trained on RGB lighting variations, RoboMD can reliably infer outcomes for novel lighting conditions, though it cannot extrapolate to fundamentally different settings (e.g., new bimanual tasks) without signals from that specific environment. To support this, we present additional results on new real-world environments, demonstrating RoboMD’s ability to adapt and capture meaningful success–failure variations across diverse physical setups.
>
> Plots for real world failure modes are given here: [Image Link](https://anonymous.4open.science/r/ICLR-Rebuttal-0A27/SeQx/Images/real_robot_policy_side_by_side.png).
>
> ## Concern 3: Definition of "Environmental Variation"
> Concrete examples of "Environmental Variation" are summarized in Appendix Figure 19, we agree that this connection could be made clearer. In the revised version, we will explicitly include this definition after the MDP formulation section to better illustrate how environmental variations correspond to actions in the semantic state space. This clarification will make the relationship between physical changes (e.g., lighting, color, geometry) and their representation in the embedding space more transparent and easier to follow.
>
> In real-world Ur5e and SO101 robot trials, we went beyond simple variations and actively introduced distractors and new table styles to increase complexity. The set of variations in the paper was deliberately chosen to cover both typical and representative real-world changes that cause failures.

---

> ### Author Response · Authors · 2025-11-22
> **Response to reviewer's comments part 3/3**
>
> ## Concern 4: Predetermined vs Virtual Rollouts
> We appreciate the request for clarification. In our experiments:
> 1. Predetermined rollouts: While higher the better, we start with a small set of known failures (typically 15-20) collected through initial testing.
> 2. Virtual rollouts: RoboMD performs exploration in embedding space, generating thousands of virtual rollouts (typically 1000-5000) without physical execution. The training process continues until the reward plateaus i.e., when the average episode reward stabilizes.
> 3. Validation rollouts: A subset of discovered vulnerabilities are validated through real-world rollouts.
>
> ## Concern 5: Spatial Understanding Limitations of VLMs
> We thank the reviewer for this observation. While earlier VLMs had limited spatial and geometric reasoning mainly due to they were not trained with such data, recent models such as Gemini Robotics-ER 1.5 [6] demonstrate strong spatial understanding for tasks like even precise object localization and 3D reasoning. RoboMD is modular by design, allowing its backbone to be seamlessly upgraded as VLM embeddings continue to improve, further enhancing spatial reasoning and failure detection capabilities. Empirically, even with a ViT+CLIP-based backbone, RoboMD already achieves strong empirical performance,
>
> ## Concern 6: Theoretical Coverage
> 1. Theorem 1: New potential field-based MDP = Naïve sparse reward MDP
> 2. Theorem 2: But the new potential field-based MDP accelerates convergence
> 3. Theorem 3: Explores success-failure boundaries efficiently
>
> The three theorems are primarily intended to establish that RoboMD’s formulation as a potential-shaped MDP leads to provably efficient and stable exploration which is a necessary foundation for covering the vulnerability space. While they do not directly model semantic completeness, they formalize the exploration dynamics that allow the policy to systematically traverse semantically meaningful regions in the embedding. Since we consider cases where there are no dynamics models (e.g., we test 450 million parameter transformers), we do not believe it is possible to provide guarantees on the coverage, akin to what we do in control backward reachability/HJB methods. However, we empirically verify results that show that this shaped exploration indeed leads to broader and more diverse vulnerability discovery.
>
> ## Questions
>
> **Q2: What is the tradeoff in predetermined examples versus virtual samples?**
>
> This is a critical question for practical deployment. The tradeoff is:
> 1. Cost: Predetermined examples require physical rollouts (expensive), while virtual samples are computationally cheap.
> 2. Coverage: Virtual samples can explore a much larger space, but may include unrealistic configurations.
>
> **Q3: What further experiments can be made to compare with other control safety methods?**
> As noted in the paper, we already compare RoboMD against several related paradigms, including RL variants (PPO, A2C, SAC).
>
> To extend this comparison, one could consider classical control safety methods such as barrier function–based approaches, which offer formal safety guarantees when system dynamics and geometries are explicitly known. However, in our case, we do not have access to system dynamics or analytic models (i.e., we do not have explicit dynamics and are working with transformer models with 450 million plus parameters).
>
> Therefore, RoboMD takes a complementary empirical perspective, learning safety behavior directly from perceptual and semantic inputs rather than relying on explicit dynamic models or handcrafted safety constraints.

---

> > ### Author Response · Authors · 2025-11-26
> >
> > We thank the reviewer again for all the insightful reviews which pushed us to strengthen the work. We truly hope we have answered the reviewers concerns regarding "Adequate Baselines" and "Safety Comparisons" by including uncertainty and conformal prediction baselines and discussing the relationship with Control Barrier Functions.
> >
> > With the discussion period ending soon, we hope that these additions address the reviewer’s concerns. If the reviewer has no further questions, we would greatly appreciate it if they might consider raising their score. We thank the reviewer for their time and consideration and are happy to clarify any remaining points.

---

### Official Review · Reviewer_TeCF · 2025-10-31

**Soundness:** 3
**Presentation:** 3
**Contribution:** 3
**Rating:** 6
**Confidence:** 4

**Summary:**

This paper proposes RoboMD, a framework for diagnosing vulnerabilities in pre-trained robot manipulation policies by recasting failure discovery as a reinforcement-learning (RL) search problem in a continuous vision–language embedding space. Instead of manually probing variations (e.g., lighting, object color, geometry), the authors train a deep RL policy (πᴹᴰ) to navigate a semantic potential field learned from limited success/failure rollouts. This field is shaped through a multimodal embedding (ViT + CLIP + MLP) trained jointly with BCE + contrastive loss. The policy explores this embedding using PPO to identify regions that correspond to likely failures and produces a failure-likelihood map.

**Strengths:**

1. I really like the idea of reframing failure discovery as an active exploration problem rather than a passive evaluation or stress test. It’s an elegant conceptual step that connects representation learning, uncertainty estimation, and RL in a unified way.
2. Extensive experiments across simulated and physical domains, ablation on embedding design (BCE vs. BCE + contrastive), and real robot validation strengthen the empirical credibility.
3. I appreciate that the paper doesn’t just stay empirical—it also provides theoretical justification via potential-based reward shaping and convergence proofs. Even if some of it is idealized, the effort to link the math with the algorithm is commendable.

**Weaknesses:**

1. The method heavily relies on the assumption that the learned multimodal embedding forms a smooth manifold where semantic similarity aligns with failure likelihood. While visualizations suggest separability, it’s unclear whether embedding-space distances correspond to physically meaningful variations. Without empirical measures of “semantic continuity,” this assumption could fail for unseen scenarios.

2. The baselines include standard RL and VLM models, but exclude other uncertainty-based failure detection or OOD methods (ensembles, energy-based models, MC-Dropout). These would provide a fairer picture of how much RoboMD really advances the state of the art in model failure analysis.

3. PPO over a 512-D continuous space is expensive and potentially unstable. The paper doesn’t analyze training cost, convergence time, or scaling to multi-object or long-horizon tasks. It’s unclear whether the approach generalizes beyond relatively simple tabletop manipulation.

4. The reported “23 % more vulnerabilities” result isn’t clearly defined. What constitutes a unique failure? A cluster in embedding space, or a distinct environmental configuration? The interpretation of this metric should be clarified.

5. While the paper demonstrates that RoboMD can generalize to unseen environment variations within the same domain, it remains unclear whether the diagnosed vulnerabilities or fine-tuned policies transfer across datasets or task families. For example, if a policy fine-tuned using RoboMD-diagnosed failures on RoboSuite were evaluated on a different benchmark such as LIBERDO or ManiSkill, would it retain improved robustness? Testing such cross-domain transfer would provide stronger evidence that RoboMD captures more fundamental, domain-invariant failure structures rather than merely overfitting to intra-domain variations.

**Questions:**

1. Table 5 reports that fine-tuning with RoboMD-guided failures outperforms fine-tuning with all failure cases, despite using a much smaller dataset (1.3 GB vs 9 GB as shown in Table 11). This is impressive but counterintuitive. Is the gain purely due to better sample efficiency (targeted failures), or does RoboMD generate more diverse failure cases in embedding space? Some discussion of this discrepancy would clarify the underlying mechanism.

2. Could the authors clarify how an action taken in the embedding space (a ∈ ℝ⁵¹²) translates into a physical environment change? Is there a learned or rule-based decoder that converts latent vectors into concrete variations (e.g., lighting, color, geometry)? If not, how are the “new failure cases” in Table 5 actually generated and validated on the robot?

3. How does RoboMD handle unseen actions or rollouts that lie outside the support of the training distribution?

---

> ### Author Response · Authors · 2025-11-21
> **Response to reviewer's comments part 1/5**
>
> We thank the reviewer for their thoughtful and detailed evaluation. We greatly appreciate their positive remarks on the conceptual novelty, empirical breadth, and theoretical grounding of our work. We are particularly encouraged that the reviewer found our idea of reframing failure discovery as an active exploration problem elegant and for recognizing the strength of our experimental validation.
>
> Please refer to this [Anonymized GitHub Link](https://anonymous.4open.science/r/ICLR-Rebuttal-0A27/TeCF/Reviewer_TeCF.md) where we have compiled detailed explanation and images/plots for better understanding.
>
> Below we address all concerns raised:
> 1. **Concern 1** - Semantic Continuity: We show new quantitative analyses showing correlation between physical semantic variations and embedding distances (τ = 1.0, r > 0.9).
> 2. **Concern 2** - Missing Baselines: We discuss how RoboMD represents a different paradigm (offline vulnerability discovery vs runtime detection) and empirically show that it outperforms the suggested uncertainty-based baselines (MC-Dropout, Ensemble), which achieve only 40–50 % accuracy.
> 3. **Concern 3** - Training Cost & Scalability: PPO converges in ~4–5 GPU hrs within 1000–1200 episodes; scalable to complex variations.
> 4. **Concern 4** - “23 % More Vulnerabilities”: Refers to distinct failure clusters unseen by VLM baselines.
> 5. **Concern 5** - Cross-Domain Transfer: Current results are within-domain; embedding structure suggests promising generalization, to be explored next.
> 6. **Questions**: Clarified that RoboMD’s gains stem from targeted, diverse failures; actions map to concrete environment changes via nearest-neighbor retrieval; and unseen actions are handled via reward-based confidence in continuous embedding space.
>
>
> ## Concern 1: Semantic Continuity
>
> We thank the reviewer for raising the point and would like to mention that the MSE-style loss (Eq. 1 in the paper) is known to form smooth surfaces due to L2 norm [1]. We now verify this empirically by (1) measuring how embedding distances vary under controlled physical changes, (2) evaluating continuity to ensure linear, consistent transitions across embeddings.
>
> Experimental Setup: We gradually vary physical parameters such as lighting, object color, and object size, and measure how these changes affect distances in the learned embedding space.
>
> Statistical Verification: We measured how these physical changes translate into distances within the learned embedding space. As shown in Rebuttal Tables 1.1 and 1.2 below, _embedding distances change monotonically with the degree of variation_, confirming that the embedding-space distances correspond to physically meaningful variations.
>
> **Rebuttal Table 1.1.** Cumulative distance from the initial demonstration increases monotonically with decreasing lighting intensity for the Pi0 policy.
>
> | Red Light Intensity              | Policy Rollouts | Cumulative Distance in Embedding |
> |----------------------------------|------------------|----------------------------------|
> | Red = 255 (completely red)       | Failed           | 0.00                             |
> | Red = 235                        | Failed           | 0.22                             |
> | Red = 187                        | Succeed          | 0.55                             |
> | Red = 163                        | Succeed          | 0.58                             |
> | Red = 140 (light pink)           | Succeed          | 0.67                             |
>
> Fig 1 (Links are in caption) shows Pi0 embedding visualization [Image 1](https://anonymous.4open.science/r/ICLR-Rebuttal-0A27/TeCF/Images/intensity_transition_embedding_space.png) t-SNE visualization of 5 rollouts projected into 2D embedding space. Points are colored by success rate (red = failure, green = success). [Image2](https://anonymous.4open.science/r/ICLR-Rebuttal-0A27/TeCF/Images/cumulative_distance_vs_red_intensity.png) Relationship between red light intensity and cumulative distance progression. Each demo (Demo 0–4) represents a step from high to low red intensity
>
> **Rebuttal Table 1.2.** Correlation analysis between physical variation and cumulative embedding distance.
>
> | Model    | Variations  | Kendall’s τ | Pearson’s r |
> |-----------|--------------|--------------|--------------|
> | BC Stack  | Cube size    | 1.000        | 0.924        |
> | BC Can    | Table color  | 1.000        | 0.842        |
>
> These perfect or near-perfect correlations (τ = 1.0, r > 0.84) confirm that semantic embedding distances grow smoothly and predictably with physical variation magnitude.

---

> ### Author Response · Authors · 2025-11-21
> **Response to reviewer's comments part 2/5**
>
> **Numerical Verification**: We further validate embedding quality using distances between consecutive embeddings (e_i and e_{i+1}) in each failure/success region. This assesses whether embeddings change at a sufficiently consistent rate, reflecting a smooth and predictable manifold suitable for policy learning. Small values of ||e1 – e2||_2 across transitions indicate continuity in the embedding space.
>
> **Rebuttal Table 1.3.**
> Continuity analysis (d₁[i] = embedding[i+1] – embedding[i]) across policies.
> | Policy       | Mean \|d₁ − d₂\| |
> |---------------|------------------|
> | Smol VLA [2]  | 0.267 ± 0.367    |
> | GR00T [3]     | 0.207 ± 0.397    |
> | π0 [4]      | 0.282 ± 0.359    |
> | ACT [5]       | 0.273 ± 0.364    |
>
> The normalized values across policies are consistent and confirm smooth transitions, verifying that RoboMD’s embedding manifold is both semantically structured and numerically smooth.
>
> **References**
> 1. Bottou, Léon, Frank E. Curtis, and Jorge Nocedal. "Optimization methods for large-scale machine learning." SIAM review 60.2 (2018): 223-311.
> 2. Shukor, Mustafa, et al. "Smolvla: A vision-language-action model for affordable and efficient robotics." arXiv preprint arXiv:2506.01844 (2025).
> 3. Zhao, Tony Z., et al. "Learning fine-grained bimanual manipulation with low-cost hardware." arXiv preprint arXiv:2304.13705 (2023).
> 4. Black, Kevin, et al. "π0: A vision-language-action flow model for general robot control. CoRR, abs/2410.24164, 2024. doi: 10.48550." arXiv preprint ARXIV.2410.24164.
> 5. Bjorck, Johan, et al. "Gr00t n1: An open foundation model for generalist humanoid robots." arXiv preprint arXiv:2503.14734 (2025).
>
> ## Concern 2: Missing Baselines
>
> We agree that failure has been an overused term with different definitions given in different contexts, especially in robotics. As we discussed in related work and Appendix B, **every OOD sample is not a failure or vice versa**. We show an example below:
> 1. OOD but not a failure (link to example [Image1](https://anonymous.4open.science/r/ICLR-Rebuttal-0A27/TeCF/Images/frame_final.jpg), [Image2](http://anonymous.4open.science/r/ICLR-Rebuttal-0A27/TeCF/Images/frame_0042.jpg))
> 2. Failure but not a OOD (link to example [Image1](https://anonymous.4open.science/r/ICLR-Rebuttal-0A27/TeCF/Images/frame_0080.jpg), [Image2](https://anonymous.4open.science/r/ICLR-Rebuttal-0A27/TeCF/Images/frame_0063.jpg))
>
> That is why we thought it might not be fair to compare with OOD detectors (now we do as per reviewer’s suggestion).
> We see three ways to detect failures:
> 1. Method 1: Building a likelihood of success with only successful trials [6] - when an unfamiliar input comes, this method will flag it. As [6] nicely shows, this is useful in runtime monitoring so that the human can intervene.
> 2. Method 2: Building a classifier with success and failure data - Method 1 is a subset of this. We have included results for this naive approach in Rebuttal Table 1.4.2. As the reviewer has requested, we have now included results of methods 1 and 2 by incorporating uncertainty as well as shown in Rebuttal Table 1.4.1 and 1.4.2.
> 3. Method 3: Searching for failures - In our method, we get an embedding from method 2 and search for failures further using deep RL. Incorporating uncertainty into method 2 and employing Bayesian RL could, in principle, further improve performance beyond our approach, and we encourage future work to explore this direction.
>
> **Rebuttal Table 1.4.1.**
> Accuracy report of models trained with only success data and thresholding (Concern 2 method 1).
>
> | Task   | Our Best | MC Dropout (0.5) CNN | MC Dropout (0.5) ResNet | Ensemble CNN | Ensemble ResNet | Ensemble CNN+ResNet |
> |--------|-----------|------------------------|---------------------------|----------------|------------------|----------------------|
> | Lift   | 82.30     | 46.95                 | 37.17                    | 45.31          | 36.10            | 37.35               |
> | Can    | 84.00     | 48.27                 | 46.99                    | 45.62          | 43.13            | 41.18               |
> | Square | 76.00     | 50.22                 | 46.30                    | 49.32          | 49.62            | 50.57               |
>
> **Rebuttal Table 1.4.2.**
> Accuracy report of models trained with both success and training data (Concern 2 method 2).
>
> | Task   | CNN | ResNet | MC Dropout (0.5) CNN | MC Dropout (0.5) ResNet | Ensemble CNN | Ensemble ResNet |
> |--------|------|---------|------------------------|---------------------------|----------------|------------------|
> | Lift   | 32.72 | 51.17  | 38.91                 | 51.23                    | 36.76          | 55.00            |
> | Can    | 35.44 | 52.25  | 37.94                 | 53.32                    | 35.27          | 48.82            |
> | Square | 14.00 | 47.00  | 21.16                 | 48.22                    | 12.92          | 43.95            |

---

> ### Author Response · Authors · 2025-11-21
> **Response to reviewer's comments part 3/5**
>
> **Rebuttal Table 1.6.3.**
> Accuracy report of models trained with Conformal Prediction and Stochastic Variational Inference.
>
> | Task   | Conformal Prediction CNN | Conformal Prediction ResNet | Stochastic Variational Inference CNN | Stochastic Variational Inference ResNet (Energy Models) |
> |--------|----------------------------|------------------------------|---------------------------------------|------------------------------------------|
> | Lift   | 25.15                     | 58.09                        | 50.88                                 | 41.97                                    |
> | Can    | 30.65                     | 55.37                        | 47.82                                 | 37.67                                    |
> | Square | 9.97                      | 53.28                        | 48.28                                 | 49.14                                    |
>
> We would also like to clarify that conceptually, RoboMD differs from the said baselines as follows:
> 1. Paradigm difference: Uncertainty-based methods perform runtime failure detection during policy execution, while RoboMD performs offline vulnerability discovery before deployment. These are complementary but serve different purposes.
> 2. Integration potential: Uncertainty-based methods could be integrated with RoboMD for enhanced performance, uncertainty estimates could guide exploration, and RoboMD-discovered vulnerabilities could improve uncertainty calibration.
>
> **References**
>
> 6. Can We Detect Failures Without Failure Data? Uncertainty-Aware Runtime Failure Detection for Imitation Learning Policies Chen Xu etc. CoRL 2024
>
> ## Concern 3: Training Cost and Scalability
>
> We thank the reviewer for raising this important practical concern. We would like to mention that our 512-D embedding space is actually modest compared to standard extremely noisy human feedback in RLHF (Reinforcement Learning from Human Feedback) applications, which successfully apply PPO in significantly higher dimensions (1000+ D) [7,8,9].
>
> Detailed analysis of training is as given below:
>
> Training Efficiency Analysis:
> 1. Computational cost: PPO training in 512-D embedding space requires ~4-5 GPU hours per task.
> 2. Convergence: PPO consistently converges within 1000-1200 episodes across all tasks, indicating high computational efficiency.
> 3. Stability: We observed stable training across all experiments with no divergence issues. As shown in Table 1 of the paper, PPO consistently outperformed SAC and A2C.
> 4. Scalability to long-horizon: Our method maps a sequence of images to failure probabilities. If a failure-causing change occurs in the environment, the failure would be predicted irrespective of the time horizon (no method, in general, can predict failures before any such signal). Hence, we consider our method to be scalable, as evidenced by its performance on the relatively complex threading task and its use of state-of-the-art VLAs.
>
> **References**
>
> 7. Ouyang, Long, et al. "Training language models to follow instructions with human feedback." Advances in neural information processing systems 35 (2022): 27730-27744.
> 8. Bai, Yuntao, et al. "Constitutional ai: Harmlessness from ai feedback." arXiv preprint arXiv:2212.08073 (2022).
> 9. Touvron, Hugo, et al. "Llama 2: Open foundation and fine-tuned chat models." arXiv preprint arXiv:2307.09288 (2023).
>
>
> ## Concern 4: Definition of "23% More Vulnerabilities”
>
> The 23% more vulnerabilities comes from RoboMD discovering failure modes that were not identified by baseline methods (GPT-4o, Gemini, Qwen2-VL) when given the same initial failure set.
>
> ## Concern 5: Cross-Domain Transfer
>
> This is an excellent point. While our current experiments focus on within-domain generalization, cross-domain transfer is an important direction for future work. However, we note that:
> 1. Embedding space structure: Our embedding space captures semantic variations (lighting, object properties, etc.) that are domain-agnostic to some extent.
> 2. Future work: We plan to investigate cross-domain transfer in future work, particularly between different manipulation tasks and robot platforms. We hope the reviewer kindly acknowledges that developing even a cross-domain manipulation policy is, as of today, an extremely challenging task as even minor variations in camera placement can lead to significant degradation in the performance of existing manipulation policies.
>
> We will add a discussion of cross-domain transfer limitations and future directions in the revised paper.

---

> ### Author Response · Authors · 2025-11-21
> **Response to reviewer's comments part 4/5**
>
> **Question 1: Table 5 - Why does fine-tuning with RoboMD-guided failures outperform fine-tuning with all failure cases despite using a smaller dataset?**
>
> This is an excellent question that gets to the heart of RoboMD's effectiveness. The improvement comes from targeted diversity (informed sampling) rather than uniform sampling as when data diversity is very high a proportionally larger amount of data is required to prevent the diverse data from acting as noise. This observation aligns with recent findings in robot learning: prior work [10, 11] has demonstrated that the informativeness and relevance of data, rather than its sheer quantity, are the primary drivers of downstream performance gains.
>
> From a learning theory perspective, even high-capacity models (e.g., the smallest Pi0 with 470M parameters) can be sensitive to overly diverse data, which may introduce noise. Although we attempted to formalize this, existing Rademacher complexity frameworks do not yet extend cleanly to modern deep learning settings.
>
> **Reference**
>
> 10. Hu, Yingdong, et al. "Data scaling laws in imitation learning for robotic manipulation." International Conference on Learning Representations (2025).
> 11. Dasari, Sudeep, et al. "Robonet: Large-scale multi-robot learning." Conference on Robot Learning (2019).
>
> **Question 2: How does an action in embedding space translate into a physical environment change?**
>
> Our method maps physical environment changes to actions in the embedding space, rather than explicitly decoding actions from it. Solving this inverse problem i.e translating arbitrary latent vectors back to precise environment configurations is often tedious. Instead, we approximate this by correlating each action with its nearest embedding neighbor based on distance (Eq. 1), which corresponds to a known configuration (e.g., lighting, color, or geometry). As future work, we plan to explore training a decoding network that can directly generate or reconstruct physical variations from arbitrary embedding-space actions.
>
> **Q3: How does RoboMD handle unseen actions or rollouts outside the training distribution?**
>
> This is an excellent question about the generalization capabilities of RoboMD. Importantly, we developed RoboMD to handle unseen actions through its embedding-space formulation:
>
> 1. Embedding-space constraint enables universal checking: Since RoboMD operates entirely in embedding space, any action can be represented in the embedding space, regardless of whether it was seen during training. The embedding space provides a continuous representation that naturally handles unseen configurations.
> 2. Reward-based confidence estimation: The reward function in RoboMD is inversely proportional to the distance from known failure modes in embedding space. Specifically:
>    - High reward (low distance): High confidence that the action corresponds to a failure mode
>    - Low reward (high distance): Low confidence, indicating the action is far from known failures
>
> If an action lies extremely far from all known regions, its prediction becomes highly uncertain,  but the good news is that this uncertainty can be quantitatively estimated by the same embedding distance,

---

> ### Author Response · Authors · 2025-11-21
> **Response to reviewer's comments part 5/5**
>
> ## Rebuttal Appendix
>
> We also test embedding quality in 4 ways: 1) we check if similar embeddings are found within well-defined success/failure regions, 2) demonstrating smooth transitions between regions, 3) confirming alignment with semantic–visual features, and 4) validating monotonic consistency with embedding variations.
>
> The quality of embeddings within each region (actions/dotted lines in Fig 4 in the paper) across different manipulation policies and tasks (which now includes more real world experiments with SOTA VLAs) were tested as follows.
>
> **Rebuttal Table 1.5.**
> Embedding-space grouping metrics showing strong separation and smoothness across tasks and policies, validating the embedding’s semantic consistency.
>
> | Domain       | Policy        | Separation Ratio (>1.5 indicates good) | Effect Size (Cohen's d) (>0.8 indicates good) | Silhouette Score (>0.5 indicates good grouping) | Davis–Bouldin Index (<1.0 indicates good) |
> |---------------|----------------|-----------------------------------------|-----------------------------------------------|--------------------------------------------------|---------------------------------------------|
> | Simulation    | BC – Lift       | 22.993                                  | 2.920                                         | 0.907                                            | 0.149                                       |
> | Simulation    | BC – Square     | 20.232                                  | 1.922                                         | 0.875                                            | 0.245                                       |
> | Real Robot    | SmolVLA [2]     | 10.371                                  | 4.522                                         | 0.871                                            | 0.347                                       |
> | Real Robot    | ACT [3]         | 7.760                                   | 4.767                                         | 0.839                                            | 0.384                                       |
> | Real Robot    | Pi0 [4]         | 23.974                                  | 4.212                                         | 0.953                                            | 0.068                                       |
> | Real Robot    | GROOT [5]       | 4.261                                   | 3.309                                         | 0.701                                            | 0.533                                       |
>
> The metrics and interpretations of Rebuttal Table 1.5 is as follows:
> 1. Separation Ratio: Computed as the ratio of mean inter-region distance to mean intra-region distance. All tasks exceed this threshold (range: 4.26-23.97), meaning failure modes form distinct, well-separated regions in embedding space.
> 2. Effect Size: Measures the standardized difference between regions relative to pooled standard deviation. All tasks show large effects (range: 1.92-4.77), demonstrating significant separation between failure mode regions.
> 3. Silhouette Score: Measures how similar an embedding is to its own region compared to other regions. All tasks exceed 0.70 (range: 0.70-0.95), demonstrating strong cohesion.
> 4. Davis-Bouldin Score: Ratio of within-region scatter to between-region separation. Lower values indicate better separation. All tasks are below 0.53 (range: 0.068-0.533), confirming excellent separation.
>
> To test the smoothness of the embedding between regions (From paper Fig 4 dotted lines), we evaluate two key properties: (1) small perturbations to known embeddings whether interpolated embeddings remain local to the region, and (2) whether embedding distances align with semantic-visual feature distances measured as cross modal action alignment.
>
> **Rebuttal Table 1.6:** Quantitative evaluation of embedding-space continuity across policies.
> | Policy   | Local Smoothness (penalty up to 15) | Cross-Modal Action Alignment |
> |-----------|-------------------------------------|-------------------------------|
> | ACT       | 93.4%                              | 58.9%                         |
> | Smol VLA  | 92.9%                              | 57.0%                         |
> | π0      | 94.3%                              | 57.9%                         |
>
> The metric and interpretation of Table 1.6 is as follows:
> 1. Local Smoothness: We add noise to each embedding and check if it still maps to small failure mode. Local Smoothness > 80% indicates very smooth local structure.
> 2. Cross-Modal Action Alignment: Computed as cosine similarity between embedding distance vectors and semantic-visual distance vectors (CLIP of Image+Text) across action pairs. Values of 55-60% indicate moderate positive correlation. This is a sweet spot because:
>     - Perfect correlation (100%) would mean the embedding adds no task specific information beyond generic semantic features.
>     - Zero correlation (0%) would mean the embedding captures no semantic meaning at all.

---

> ### Author Response · Authors · 2025-11-26
>
> We truly hope we have answered all of the reviewer's concerns by providing new quantitative correlation analysis (τ=1.0, r>0.8), comparing against uncertainty-based methods and real-world experiments on SOTA VLAs.
>
> If our response and new data have resolved the reviewer's doubts, we would be grateful if you could consider raising your rating. Since the rebuttal period is ending soon, kindly let us know if there are any other clarifications we can provide. Once again, thank you for supporting our research.

---

### Meta-Review · Area_Chair_kzJK · 2026-01-13

**Summary:**

This paper proposes to learn a separate deep reinforcement learning (deep RL) policy for vulnerability prediction through virtual runs on a continuous vision-language embedding trained with limited success-failure data. Basically, the initial reviews from 3/4 reviewers are positive. Reviewer 5NX2 has some concerns about the technique details and evaluation in the physical world, as well as some comparison issues. All reviewers confirm the novelty of this paper in different statements, making this paper more likely to be accepted. During the discussion period, the authors provide careful response according to the issues from reviewers, which might be able to address all concerns in my opinion. Thus, I lean to accept this paper.

**Reviewer Concerns:**

The reviewers acknowledge the practical relevance of the work but raise substantial methodological, empirical, and presentational concerns that must be addressed to establish its significance and validity. The major concerns lies on the methodological foundations and assumptions, inadequate and unfair empirical comparisons, limited experimental scope and validation, clarity of metrics and contributions. Basicially, the authors make great efforts on addressing the mentioned issues by a point2point answering. Thus, I do not think that there are concerns unaddressed.

**Reviewer Scores:**

Reviewer DxbL clearly decided to maintain the score. And other reviewers did not reply. After carefully reading the response from authors, I think Reviewer 5NX2 might upgrade the rating, and the other reviewers might maintain the ratings at least.

---

### Decision · Program_Chairs · 2026-01-26

Accept (Poster)